# YAP promotes the activation of NLRP3 inflammasome via blocking K27-linked polyubiquitination of NLRP3

Dan Wang[1,2,10], Yening Zhang[1,10], Xueming Xu[1], Jianfeng Wu[3], Yue Peng [4], Jing Li[1], Ruiheng Luo[1], Lingmin Huang[1], Liping Liu [5], Songlin Yu[1,6], Ningjie Zhang[7], Ben Lu [1,8,9 ✉] & Kai Zhao [1 ✉]

The transcription coactivator YAP plays a vital role in Hippo pathway for organ-size control and tissue homeostasis. Recent studies have demonstrated YAP is closely related to immune disorders and inflammatory diseases, but the underlying mechanisms remain less defined. Here, we find that YAP promotes the activation of NLRP3 inflammasome, an intracellular multi-protein complex that orchestrates host immune responses to infections or sterile injuries. YAP deficiency in myeloid cells significantly attenuates LPS-induced systemic inflammation and monosodium urate (MSU) crystals-induced peritonitis. Mechanistically, YAP physically interacts with NLRP3 and maintains the stability of NLRP3 through blocking the association between NLRP3 and the E3 ligase β-TrCP1, the latter increases the proteasomal degradation of NLRP3 via K27-linked ubiquitination at lys380. Together, these findings establish a role of YAP in the activation of NLRP3 inflammasome, and provide potential therapeutic target to treat the NLRP3 inflammasome-related diseases.

[1] Department of Hematology and Key Laboratory of Non-resolving Inflammation and Cancer of Hunan Province, The Third Xiangya Hospital, Central South University, Changsha, Hunan Province, People's Republic of China. [2] Department of Dermatology, The Third Xiangya Hospital, Central South University, Changsha, Hunan Province, People's Republic of China. [3] State Key Laboratory of Cellular Stress Biology Innovation Center for Cell Signaling Network, School of Life Sciences, Xiamen University, Xiamen, Fujian Province, People's Republic of China. [4] Department of Critical Care Medicine, The Third Xiangya Hospital, Central South University, Changsha, Hunan Province, People's Republic of China. [5] Department of General Surgery, The Third Xiangya Hospital, Central South University, Changsha, Hunan Province, People's Republic of China. [6] Postdoctoral Research Station of Clinical Medicine, The Third Xiangya Hospital, Central South University, Changsha, Hunan Province, People's Republic of China. [7] Department of Blood Transfusion, The Second Xiangya Hospital, Central South University, Changsha, Hunan Province, People's Republic of China. [8] Department of Pathophysiology, School of Basic Medical Science, Central South University, Changsha, Hunan Province, People's Republic of China. [9] Key Laboratory of Sepsis and Translational Medicine, School of Basic Medical Science, Central South University, Changsha, Hunan Province, People's Republic of China. [10] These authors contributed equally: Dan Wang, Yening Zhang. ✉ email: xybenlu@csu.edu.cn; kaizhao@csu.edu.cn

The NLRP3 inflammasome is a multiple protein complex that plays critical role in host immune responses to infections or sterile injuries[1–3]. It consists of the nucleotide-binding domain, leucine-rich repeat, pyrin domain-containing protein 3 (NLRP3), the apoptosis-associated speck like protein containing a CARD (ASC) and caspase-1. NLRP3 can be activated by a broad range of microbial components, endogenous danger signals and environmental irritants[1]. Once activated, the NLRP3 inflammasome serves as a platform for the activation of caspase-1, which induces the maturation of interleukin (IL)-1β and IL-18, as well as gasdermin D-mediated pyroptotic cell death[1–3]. It is well-known that the deregulated NLRP3 inflammasome drives the progression of many inflammatory, metabolic, degenerative and aging-related diseases, such as atherosclerosis, gout, type 2 diabetes, autoimmune disorders, and Alzheimer's disease[4–7]. Therefore, NLRP3 inflammasome activity must be tightly regulated to maintain immune homeostasis and avoid detrimental effects. Although the mechanisms by which NLRP3 inflammasome is activated has been studied extensively, its endogenous regulatory networks remain largely unknown.

The Hippo-YAP pathway, which is highly evolutionarily conserved in mammals, is a key regulator to control the organ size[8,9]. This pathway integrates a variety of cellular stress signaling to maintain tissue homeostasis, including mechanical stress, DNA damage and oxidative stress under physiological and pathological conditions[10]. Upon activation of the Hippo pathway, the transcription co-activators YAP and TAZ are phosphorylated by Lats1/2 and retained in the cytoplasm for degradation. When the signaling is turned off, YAP/TAZ are predominantly localized in the nucleus, binding to transcription factors of the TEAD family and other factors to promote gene transcription[8,9]. Recent studies have revealed the role of the Hippo-YAP pathway in innate antibacterial[11–13] and antiviral immune responses[14,15] via transcriptional or non-transcriptional activity, providing new insights into the crosstalk between the innate immune system and the Hippo-YAP pathway. Moreover, the elevated expression of YAP is associated with the progression of atherosclerosis[16], inflammatory bowel disease (IBD)[17], sepsis[18], and pancreatitis[19]. Together with the findings that these inflammatory disorders involve the deregulated NLRP3 inflammasome activity, we postulated that YAP might regulates the activation of NLRP3 inflammasome.

In this work, we demonstrate that YAP promotes the activation of NLRP3 inflammasome. Deletion of YAP in myeloid cells significantly attenuates lipopolysaccharide (LPS)-induced systemic inflammation and monosodium urate (MSU) crystals-induced peritonitis. Activation of the Hippo signaling that induces the degradation of YAP markedly decreases the activity of the NLRP3 inflammasome in vitro. Mechanistically, YAP physically interacts with NLRP3 and maintains its stability through blocking the association between NLRP3 and the E3 ligase β-TrCP1, the latter promotes the proteasomal degradation of NLRP3 via K27-linked ubiquitination at lys380. Thus, we uncover a role of YAP in regulation of the NLRP3 inflammasome activation, and provide potential therapeutic target to treat a number of inflammatory disorders, such as atherosclerosis, gout, colitis, and sepsis.

## Results

### YAP specifically promotes NLRP3 inflammasome activation.
To investigate the role of YAP in the NLRP3 inflammasome activation, we diminished the YAP expression in primary mouse macrophages using small-interfering RNA (siRNA) (Supplementary Fig. 1a). Knockdown of YAP expression markedly suppressed ATP-, nigericin-, or MSU-induced caspase-1 cleavage and release of IL-1β and IL-18, without affecting the tumor necrosis

factor alpha (TNF-α) secretion (Fig. 1a, b). We also examined whether YAP specifically regulates the NLRP3 inflammasome activation and found that silencing of YAP expression had no effect on either poly (dA:dT)-induced AIM2 inflammasome or flagellin (FLA) transfection-induced NLRC4 inflammasome activation (Fig. 1a). To confirm this phenomenon, we generated *Yap^fl/fl lyz2-Cre* mice with *Yap* deletion in myeloid cells[20]. ATP, nigericin or MSU-induced cleavage of caspase-1 and release of IL-1β and IL-18 were markedly suppressed in YAP-deficient (*Yap^fl/fl lyz2-Cre*) macrophages as compared to that of YAP-sufficient (*Yap^fl/fl*) macrophages (Fig. 1c, d). By contrast, YAP deletion did not alter the release of IL-1β and IL-18 upon poly (dA:dT) or flagellin transfection (Fig. 1c, d). The NLRP3 inflammasome activation induces the oligomerization of ASC, resulting in formation of ASC specks through the pyrin domain[1–3]. Accordingly, deletion of YAP markedly inhibited the ASC oligomerization and ASC-speck formation (Fig. 1e, f). NLRP3 is pivotal for caspase-1-dependent IL-1β maturation upon non-canonical inflammasome activation[21]. As expected, YAP deficiency suppressed both caspase-1 cleavage and IL-1β release in macrophages transfected with LPS (Supplementary Fig. 1b, c). Taken together, our data demonstrate that YAP deficiency specially impairs the NLRP3 inflammasome activation.

### Cytoplasmic YAP inhibits the proteasomal degradation of NLRP3.
Next, we sought to investigate the mechanisms by which YAP promotes the NLRP3 inflammasome activation. It is well-established that the activation of NLRP3 inflammasome requires two steps, termed priming and assembly[1]. The priming step involves the stimulation of NLRP3 expression, which is critical for the subsequent inflammasome activation[1,3]. We examined the expression of NLRP3 inflammasome components in mouse macrophages following LPS stimulation and found that YAP deficiency markedly decreased the protein expression of NLRP3 without affecting the expression of caspase-1, pro-IL-1β, and ASC (Fig. 2a). Intriguingly, deletion of YAP failed to affect the mRNA expression of NLRP3 (Supplementary Fig. 2a), clearly suggest that YAP promotes NLRP3 expression at the post-transcriptional level. In line with the previous findings that NLRP3 degradation impairs the NLRP3 inflammasome activation[1], YAP deficiency significantly promoted the protein degradation of NLRP3 but not ASC or caspase-1 (Fig. 2b, c). NLRP3 degradation was reversed by the proteasome inhibitor MG-132, but not the lysosome inhibitor chloroquine or the autophagy inhibitor 3-MA (Fig. 2d–f). suggesting that YAP inhibits the proteasomal degradation of NLRP3.

Previous studies[22,23] have reported that substitution of YAP residue Ser112 with Ala (S112A) generates a nuclear protein with constitutive transcriptional activity (Supplementary Fig. 2b). Using genetically modified HEK293T cells expressing the NLRP3 inflammasome components, we observed that overexpression of wild-type but not S112A mutant YAP markedly increased the IL-1β release (Fig. 2g) and promoted the NLRP3 expression in a dose-dependent manner (Fig. 2h), while the S112A YAP mutant had no such effect (Fig. 2g, h). Moreover, YAP predominantly localized in the cytoplasm of macrophages upon LPS stimulation (Supplementary Fig. 2c, d). These results suggest that cytoplasmic YAP is critical for maintaining the stability of NLRP3.

### Hippo signaling suppresses the NLRP3 inflammasome activation in a YAP-dependent manner.
The Hippo pathway, activated by various cellular stresses, is known to induce the degradation of YAP[8,9]. We then asked whether cellular stresses-induced Hippo signaling could affect the NLRP3 inflammasome activation, and subjected cells to serum starvation or high cell

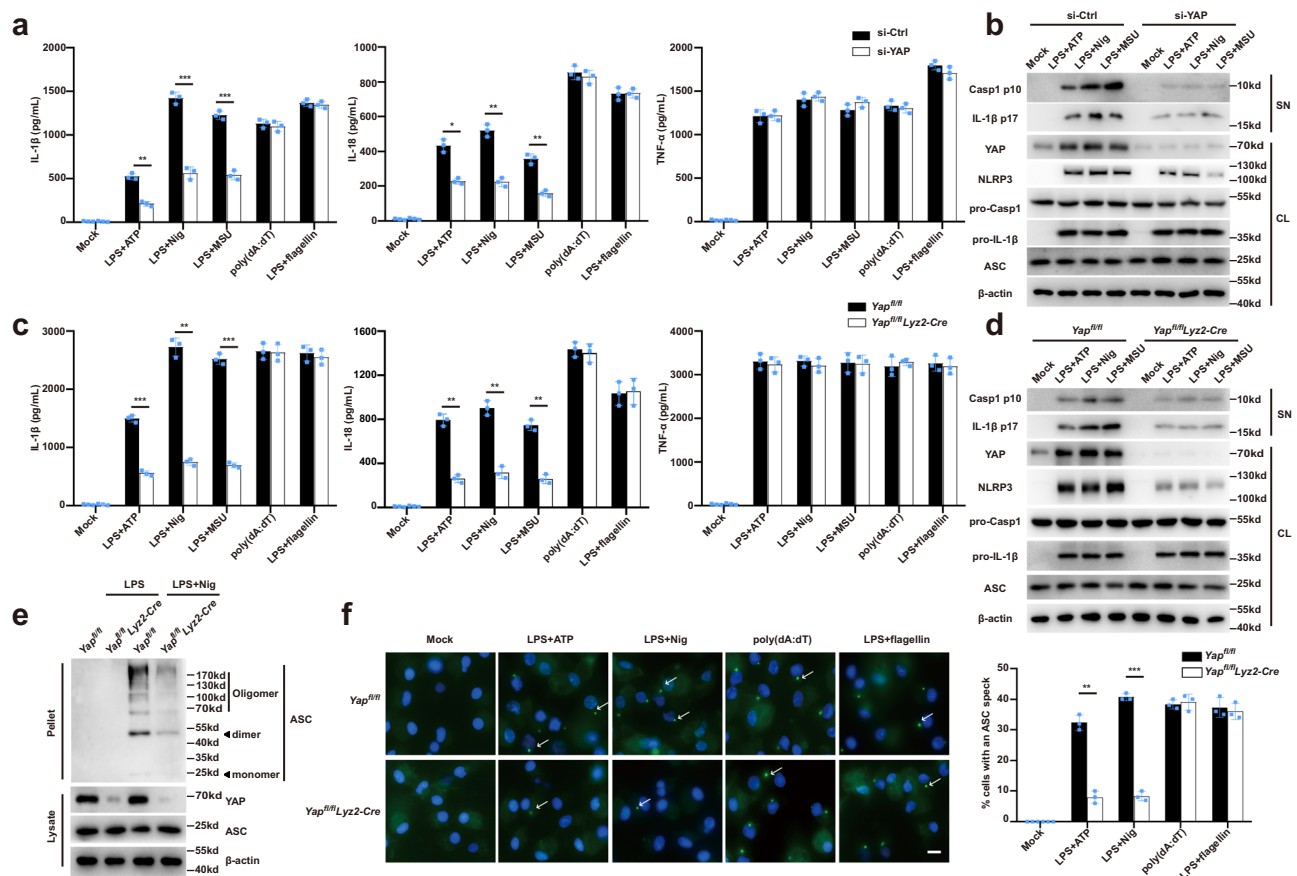

**Fig. 1 YAP specifically promotes NLRP3 inflammasome activation. a** ELISA of IL-1β, IL-18, and TNF-α in supernatants from mouse peritoneal macrophages silenced of YAP, treated with indicated stimuli (mean ± SD, two-way ANOVA with Bonferroni test, YAP siRNA vs. Ctrl siRNA, left panel: **P = 0.0023, ***P = 0.0007, ***P = 0.0005 in sequence; middle panel: *P = 0.0222, **P = 0.0033, **P = 0.0076 in sequence; n = 3 independent experiments). **b** Immunoblot analysis of supernatants (SN) or cell lysates (CL) from mouse peritoneal macrophages silenced of YAP, treated with indicated stimuli. **c** ELISA of IL-1β, IL-18, and TNF-α in supernatants of mouse peritoneal macrophages from Yap^fl/fl lyz2-Cre or Yap^fl/fl mice, then treated with indicated stimuli (mean ± SD, two-way ANOVA with Bonferroni test, Yap^fl/fl lyz2-Cre vs. Yap^fl/fl, left panel: ***P = 0.0003, **P = 0.0046, ***P = 0.0006 in sequence; middle panel: **P = 0.0021, **P = 0.002, **P = 0.0013 in sequence; n = 3 independent experiments). **d** Immunoblot analysis of supernatants (SN) or cell lysates (CL) of mouse peritoneal macrophages from Yap^fl/fl lyz2-Cre or Yap^fl/fl mice, then treated with indicated stimuli. **e** Immunoblot analysis of ASC oligomerization in cross-linked cytosolic pellets of mouse peritoneal macrophages from Yap^fl/fl lyz2-Cre or Yap^fl/fl mice, primed with LPS, and followed by stimulation with nigericin. **f** Representative images of ASC specks in LPS primed Yap^fl/fl lyz2-Cre or Yap^fl/fl peritoneal macrophages treated with indicated stimuli. ASC, green; nuclei, blue. White arrows indicate ASC specks. Scale bars, 10 μm (left). The percentage of cells containing an ASC speck was quantified (right). At least 100 peritoneal macrophages from each genotype were analyzed (mean ± SD, two-way ANOVA with Bonferroni test, Yap^fl/fl lyz2-Cre vs. Yap^fl/fl, right panel: **P = 0.0011, ***P < 0.0001 in sequence; n = 3 independent experiments). Similar results were obtained from three independent experiments. Source data are provided as a Source Data file.

confluence, both of which are well-known cellular stress conditions that activate the Hippo signaling[24,25], this led to YAP Ser127 phosphorylation and YAP degradation, indicative of the activation of Hippo signaling (Supplementary Fig. 3a, b). Serum starvation or high cell confluence markedly inhibited the NLRP3 inflammasome-induced release of IL-1β and IL-18 in WT but not YAP-deficient macrophages, without affecting the secretion of TNF-α (Fig. 3a, b). Upon activation of the Hippo signaling, Lats1/2 mediates YAP phosphorylation, leading to its polyubiquitination and degradation[24,25]. To further confirm that the Hippo signaling regulates the activation of NLRP3 inflammasome through YAP, we next investigated the role of Lats1/2 in activation of the NLRP3 inflammasome. Silencing of Lats1/2 expression in macrophages prevented the decrease of YAP expression (Supplementary Fig. 3c, d), and enhanced NLRP3 agonists-induced IL-1β and IL-18 secretion during serum starvation (Fig. 3c). The coactivator TAZ is a downstream effector of Hippo signaling[8,9]. However, in contrast to the lung tissues,

macrophages did not express detectable TAZ expression in the presence or the absence of LPS stimulation (Supplementary Fig. 3e, f). Taken together, our findings indicated that the Hippo signaling suppresses the NLRP3 inflammasome activation in a YAP-dependent manner.

**YAP suppresses K27-linked ubiquitination of NLRP3**. Protein ubiquitination is a crucial step in the ubiquitin–proteasome degradation pathway[26]. We next determined whether YAP participates in the ubiquitination of NLRP3. We found that NLRP3 ubiquitination was markedly decreased in HEK293T cells over-expressing YAP (Fig. 4a). By contrast, deletion of YAP in myeloid cells increased the NLRP3 ubiquitination (Fig. 4b). Since different types of polyubiquitin linkages mediate distinct biological functions, we expressed Myc-tagged NLRP3 in HEK293T cells together with a series of ubiquitin mutants (K6, K11, K27, K29, K33, K48, and K63), all of which contained only one indicated lysine available for

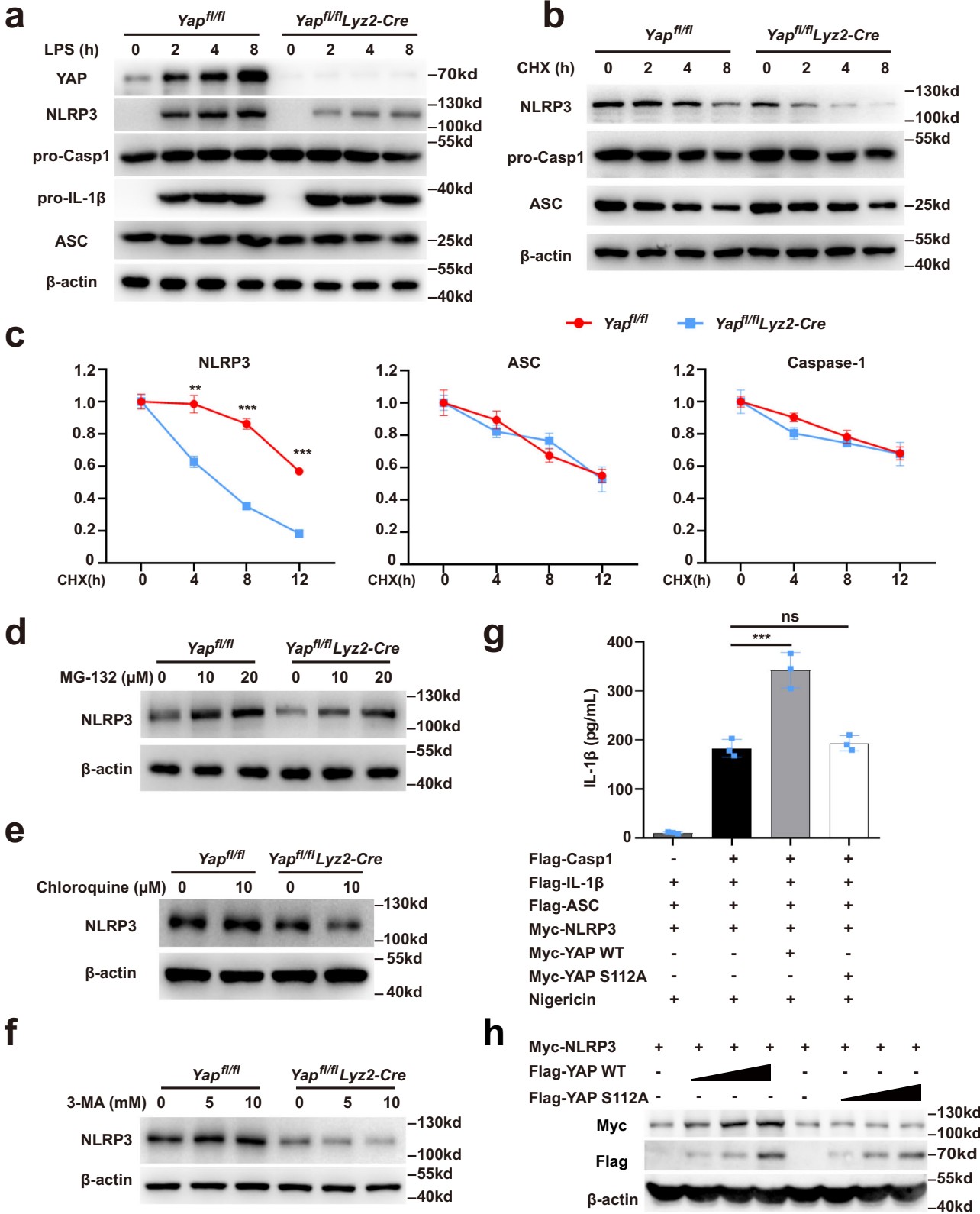

poly-linkage. Notably, over-expressing YAP markedly decreased K27-linked polyubiquitination of NLRP3 but had no appreciable effect on the NLRP3 ubiquitination of other types of linkages (Fig. 4c). When we replaced K27 with an arginine (K27R), YAP was no longer able to decrease the polyubiquitination of NLRP3 (Fig. 4d). Taken together, these data suggest that YAP selectively suppresses the K27-linked ubiquitination of NLRP3.

**YAP suppresses β-TrCP1-mediated K27-linked ubiquitination of NLRP3.** Since YAP could block the ubiquitination of NLRP3, we speculated that there may exist a YAP-associated enzyme that mediates NLRP3 ubiquitination. The 14-3-3 protein has been reported to bind to phosphorylated YAP, sequestering it in the cytoplasm[8,9]. Previous studies have suggested that 14-3-3σ promotes the ubiquitination of c-myc and COP1[27,28], prompting us

**Fig. 2 Cytoplasmic YAP inhibits the proteasomal degradation of NLRP3. a** Immunoblot analysis of extracts from *Yap^fl/fl lyz2-Cre* or *Yap^fl/fl* mouse peritoneal macrophages, then stimulated for indicated times with LPS. **b**, **c** Immunoblot analysis of extracts from *Yap^fl/fl Iyz2-Cre* or *Yap^fl/fl* mouse peritoneal macrophages stimulated with LPS for 4 h, and then treated for various times with cycloheximide (CHX) (**b**). NLRP3, ASC and Caspase-1 expression levels were quantitated by measuring band intensities using "ImageJ" software. The values were normalized to actin (**c**) (mean ± SD, two-way ANOVA with Bonferroni test, *Yap^fl/fl lyz2-Cre* vs. *Yap^fl/fl*, left panel: **P = 0.0054, ***P = 0.0002, ***P = 0.0001 in sequence; n = 3 independent experiments). **d–f** Immunoblot analysis of extracts from *Yap^fl/fl lyz2-Cre* or *Yap^fl/fl* mouse peritoneal macrophages stimulated with LPS for 4 h, and then treated with different doses of MG-132 (**d**), and chloroquine (**e**), and 3-MA (**f**) for 8 h. **g** ELISA of IL-1β in supernatants from HEK293T cells transfected with NLRP3, ASC, pro-caspase-1, pro-IL-1β, YAP-WT or YAP-S112A plasmids and stimulated with 10 μM nigericin (mean ± SD, one-way ANOVA with Bonferroni test, the third and fourth group vs. the second group ***P = 0.0001, ns > 0.9999; n = 3 independent experiments). **h** Immunoblot analysis of lysates from HEK293T cells transfected with NLRP3, YAP-WT, or YAP-S112A plasmids. Similar results were obtained from three independent experiments. Source data are provided as a Source Data file.

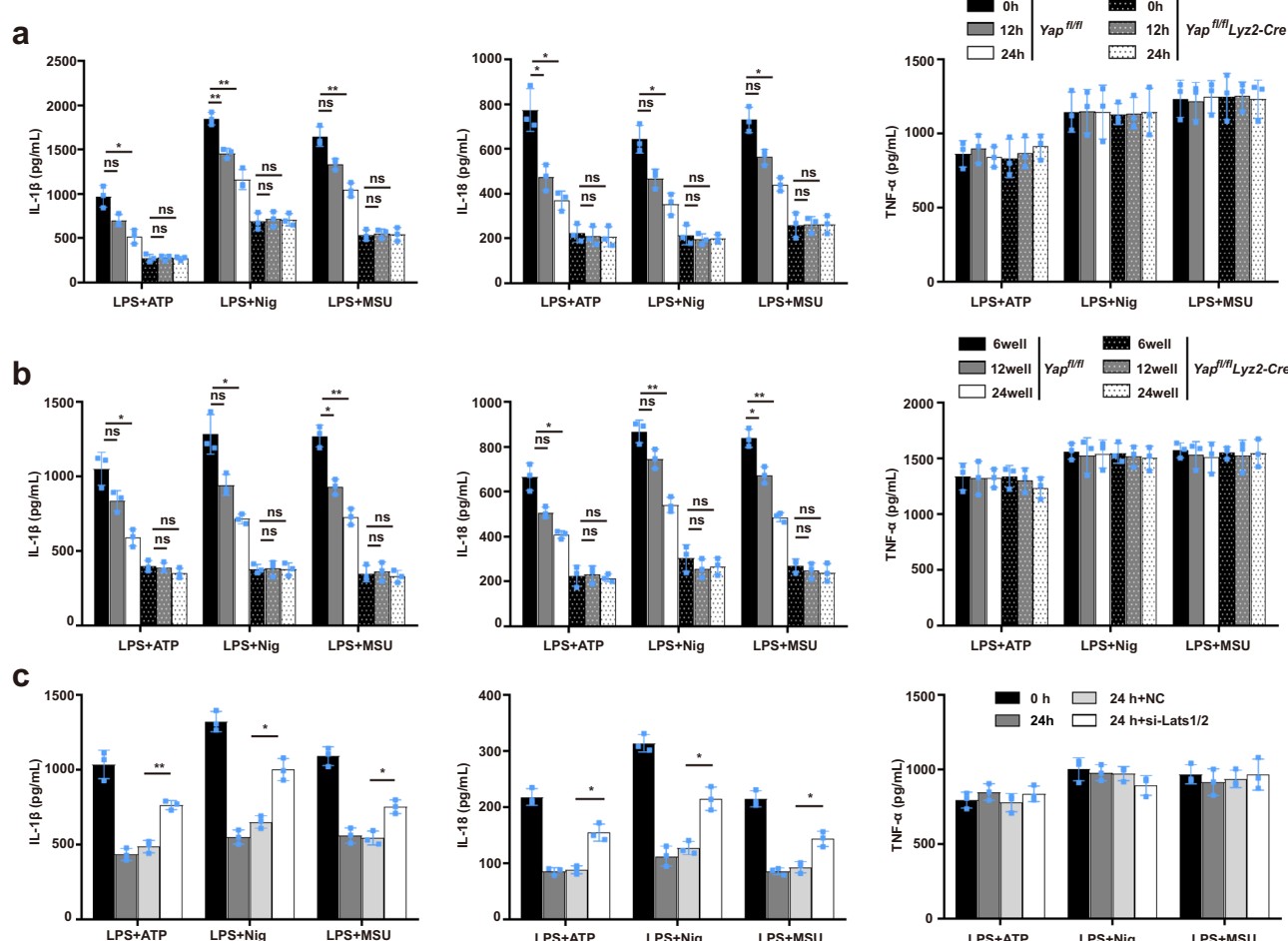

**Fig. 3 Hippo signaling suppresses the NLRP3 inflammasome activation in a YAP-dependent manner. a**, **b** ELISA of IL-1β, IL-18, and TNF-α in supernatants of mouse peritoneal macrophages from *Yap^fl/fl lyz2-Cre* and *Yap^fl/fl* mice treated with serum starvation for indicated times (**a**) or seeded into different confluence (**b**), then applied with indicated stimuli (for **a**, mean ± SD, two-way ANOVA with Bonferroni test, left panel:12 h and 24 h vs. 0 h in *Yap^fl/fl* or *Yap^fl/fl lyz2-Cre* group, ns = 0.1243, *P = 0.0267, ns > 0.9999, ns > 0.9999, **P = 0.0065, **P = 0.0055, ns > 0.9999, ns > 0.9999, ns = 0.0661, **P = 0.0069, ns > 0.9999, ns > 0.9999 in sequence; middle panel:12 h and 24 h vs. 0 h in *Yap^fl/fl* or *Yap^fl/fl lyz2-Cre* group, *P = 0.0448, *P = 0.0261, ns > 0.9999, ns > 0.9999, ns = 0.06, *P = 0.0113, ns > 0.9999, ns > 0.9999, ns = 0.0557, *P = 0.0127, ns > 0.9999, ns > 0.9999 in sequence; n = 3 independent experiments. For **b**, mean ± SD, two-way ANOVA with Bonferroni test, left panel: 12-well and 24-well vs. 6-well in *Yap^fl/fl* or *Yap^fl/fl lyz2-Cre* group, ns = 0.1743, *P = 0.0239, ns > 0.9999, ns = 0.5177, ns = 0.0861, *P = 0.0408, ns > 0.9999, ns > 0.9999, *P = 0.0115, **P = 0.0021, ns > 0.9999, ns > 0.9999 in sequence; middle panel: 12 h and 24 h vs. 0 h in *Yap^fl/fl* or *Yap^fl/fl lyz2-Cre* group, ns = 0.0969, *P = 0.0386, ns > 0.9999, ns > 0.9999, ns = 0.1, **P = 0.0038, ns > 0.9999, ns > 0.9999, *P = 0.0205, **P = 0.0032, ns > 0.9999, ns > 0.9999 in sequence; n = 3 independent experiments). **c** ELISA of IL-1β, IL-18, and TNF-α in supernatants from mouse peritoneal macrophages silenced of Lats1/2 and treated with serum starvation for indicated times, then applied with indicated stimuli (mean ± SD, two-way ANOVA with Bonferroni test, si Lats1/2 24 h vs. si Ctrl 24 h, left panel: **P = 0.0031, *P = 0.0107, *P = 0.0492 in sequence; middle panel: *P = 0.0492, *P = 0.0393, *P = 0.0478 in sequence; n = 3 independent experiments). Source data are provided as a Source Data file.

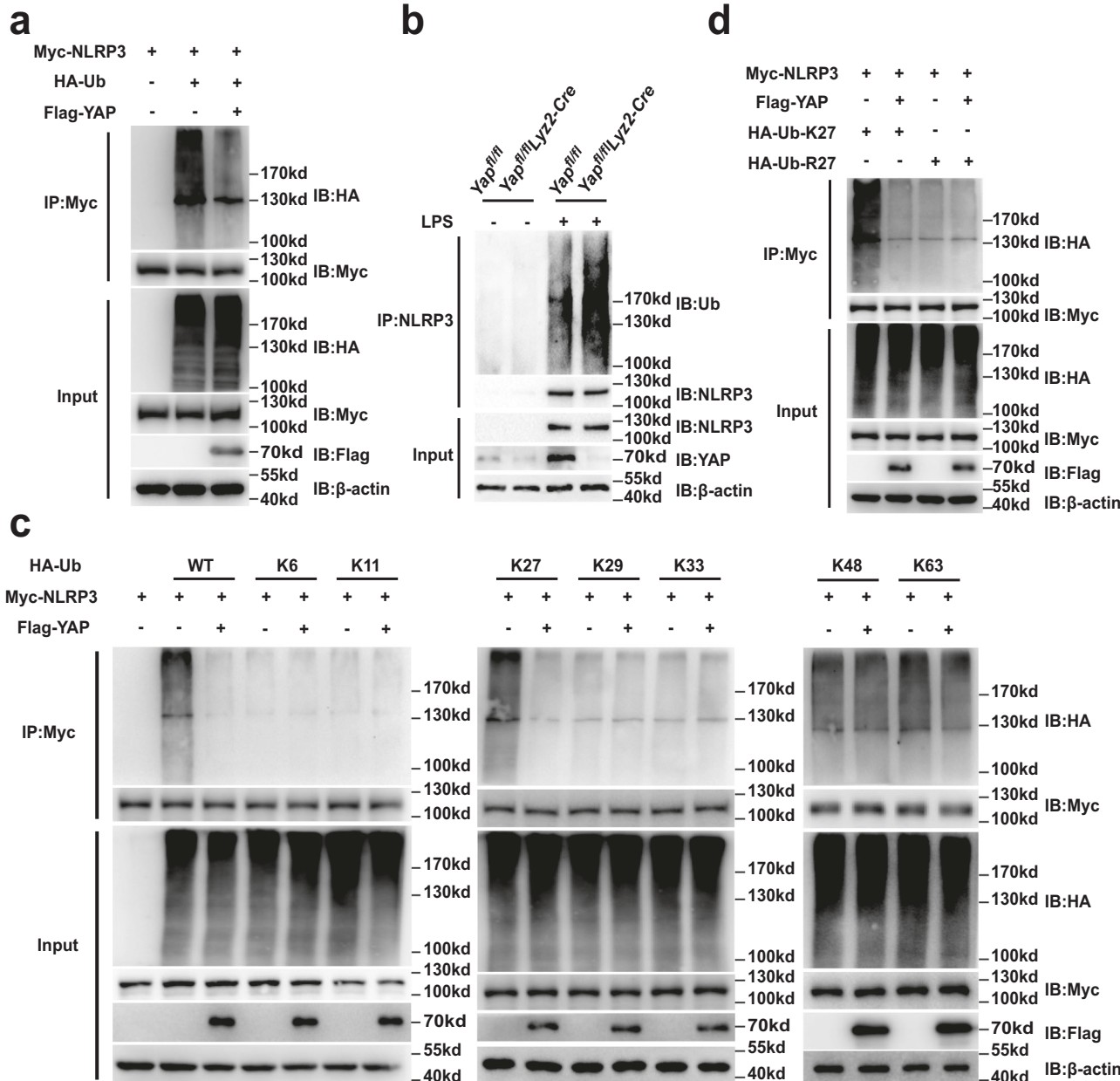

**Fig. 4 YAP suppresses K27-linked ubiquitination of NLRP3. a** Immunoblot analysis of lysates from HEK293T cells transfected with HA-tagged ubiquitin (HA-Ub), Myc-NLRP3 and Flag-YAP, followed by IP with anti-Myc, probed with anti-HA. **b** Immunoblot analysis of lysates from $Yap^{fl/fl}$ $lyz2$-Cre or $Yap^{fl/fl}$ mouse peritoneal macrophages stimulated with LPS for 4 h or not, followed by IP with anti-NLRP3, probed with anti-Ub. **c** Immunoblot analysis of lysates from HEK293T cells transfected with HA-tagged ubiquitin (HA-Ub), HA-tagged K6-linked ubiquitin (K6-Ub), HA-tagged K11-linked ubiquitin (K11-Ub), HA-tagged K27-linked ubiquitin (K27-Ub), HA-tagged K29-linked ubiquitin (K29-Ub), HA-tagged K33-linked ubiquitin (K33-Ub), HA-tagged K48-linked ubiquitin (K48-Ub) or HA-tagged K63-linked ubiquitin (K63-Ub), Myc-NLRP3 and Flag-YAP, followed by IP with anti-Myc, probed with anti-HA. **d** Immunoblot analysis of lysates from HEK293T cells transfected with HA-tagged K27-linked ubiquitin (K27-Ub) or K27R-linked ubiquitin (K27R-Ub), Myc-NLRP3 and Flag-YAP, followed by IP with anti-Myc, probed with anti-HA. Similar results were obtained from three independent experiments. Source data are provided as a Source Data file.

to examine the role of 14-3-3σ in NLRP3 ubiquitination. However, expressing 14-3-3σ did not enhance the ubiquitination of NLRP3 (Supplementary Fig. 4a).

Other studies report that β–transducin repeat containing E3 ubiquitin protein ligase 1 (β-TrCP1), targets YAP for ubiquitination and degradation[29–32]. Thus, we next determined whether β-TrCP1 could mediate the ubiquitination of NLRP3. To address this, we first measured the interaction between β-TrCP1 and NLRP3. We found that β-TrCP1 co-immunoprecipitated with NLRP3 in HEK293T cells expressing both Flag-tagged β-TrCP1

and Myc-tagged NLRP3 (Supplementary Fig. 4b). Co-immunoprecipitation assays confirmed that endogenous β-TrCP1 interacted with NLRP3 in LPS-stimulated primary macrophages (Supplementary Fig. 4c). Further, confocal microscopy revealed the colocalization of β-TrCP1 and NLRP3 in primary macrophages (Supplementary Fig. 4d). As shown by previous studies[33,34], the classic $DSG(X)_{2+N}S$ motif and "atypical" $XSG(X)_{2+N}S$ motif are important characteristics of substrates for β-TrCP1 binding. We therefore scanned the NLRP3 protein sequence and identified two potential phosphodegron motifs,

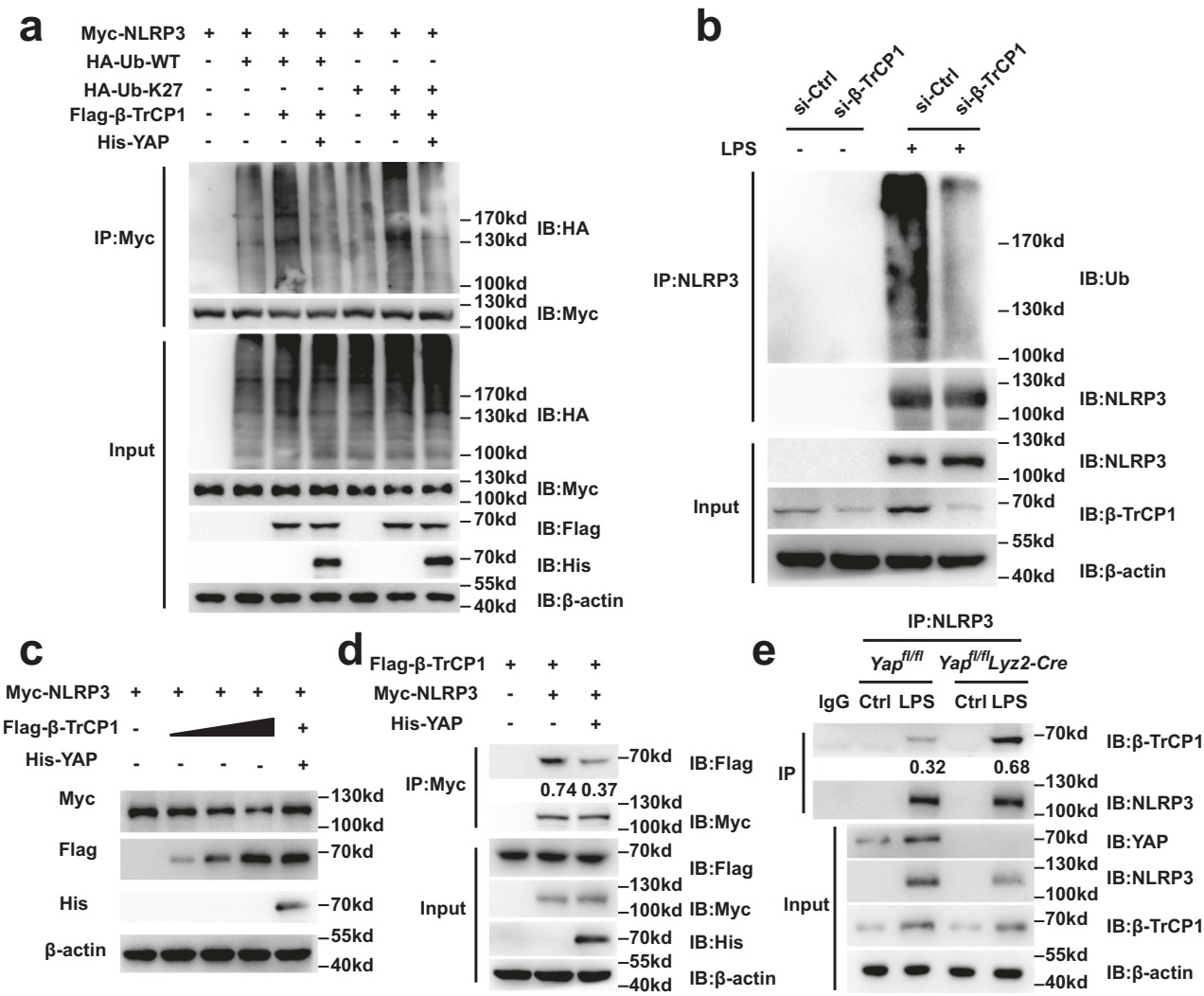

**Fig. 5 YAP suppresses β-TrCP1-mediated K27-linked ubiquitination of NLRP3. a** Immunoblot analysis of lysates from HEK293T cells transfected with HA-tagged ubiquitin (HA-Ub), HA-tagged K27-linked ubiquitin (K27-Ub), Myc-NLRP3 and Flag- β-TrCP1, with or without His-YAP, followed by IP with anti-Myc, probed with anti-HA. **b** Immunoblot analysis of cell lysates from mouse peritoneal macrophages silenced of β-TrCP1 stimulated with LPS for 4 h or not, followed by IP with anti-NLRP3, probed with anti-Ub. **c** Immunoblot analysis of lysates from HEK293T cells transfected with Myc-NLRP3, Flag- β-TrCP1, with or without His-YAP. **d** Immunoblot analysis of lysates from HEK293T cells transfected with Myc-NLRP3, Flag- β-TrCP1, with or without His-YAP, followed by IP with anti-Myc, probed with anti-Flag. Quantification of relative Flag levels was shown in the bottom panel, which was normalized to IP-Myc. **e** Immunoblot analysis of lysates from *Yapfl/fl lyz2-Cre* or *Yapfl/fl* mouse peritoneal macrophages stimulated with LPS for 4 h or not, followed by IP with anti-NLRP3, probed with anti- β-TrCP1. Quantification of relative β-TrCP1 levels is shown in the bottom panel, which was normalized to IP-NLRP3. Similar results were obtained from three independent experiments. Source data are provided as a Source Data file.

namely 193-DSPMSS-198 and 890-NSGLTS-895; the latter is conserved among species (Supplementary Fig. 4e). Given that serine phosphorylation within the degron is crucial for the recognition by β-TrCP1, we generated mutants in which all serine residues within the putative degron motifs were replaced with alanine. The NLRP3 S891A/S895A mutant lost its ability to bind to β-TrCP1, while NLRP3 S194A/S197A/S198A bound to β-TrCP1 in a manner similar to that of wild-type NLRP3 (Supplementary Fig. 4f), suggesting that the 890-NSGLTS-895 motif of NLRP3 is crucial for its binding to β-TrCP1.

Next, we tested whether β-TrCP1 mediates the ubiquitination of NLRP3. We found that the K27-linked ubiquitination and the overall polyubiquitination of NLRP3 was markedly increased in HEK293T cells overexpressing β-TrCP1 (Fig. 5a). Silencing β-TrCP1 expression in primary macrophages decreased the ubiquitination of NLRP3 (Fig. 5b). These data clearly suggest that β-TrCP1 binds to NLRP3 and mediates the ubiquitination of

NLRP3. To further investigate the relationship between YAP and β-TrCP1 in mediating NLRP3 ubiquitination, we expressed YAP, β-TrCP1, and NLRP3 in HEK293T cells and observed that YAP over-expression blocked β-TrCP1-induced polyubiquitination and K27-linked ubiquitination of NLRP3 (Fig. 5a). Accordingly, the decreased NLRP3 expression mediated by β-TrCP1 was rescued by YAP overexpression (Fig. 5c). Collectively, these data indicate that YAP blocks β-TrCP1-mediated K27-linked ubiquitination and polyubiquitination of NLRP3.

**YAP disrupts the interaction between NLRP3 and β-TrCP1 through competing with β-TrCP1 to bind NLRP3.** To explore the mechanism by which YAP blocks β-TrCP1-mediated K27-linked ubiquitination of NLRP3, we determined whether YAP affects the physical interaction between β-TrCP1 and NLRP3. Co-expression of YAP decreased the interaction between β-TrCP1 and NLRP3 (Fig. 5d), whereas YAP deficiency increased the

capacity of β-TrCP1 to bind NLRP3 (Fig. 5e), indicating that YAP disrupts the interaction between β-TrCP1 and NLRP3. Three possible explanations might explain the underlying mechanisms. First, YAP may affect the serine phosphorylation of NLRP3, leading to the decreased binding between NLRP3 and β-TrCP1. Second, YAP might directly interact with β-TrCP1 and thereby mask the NLRP3 binding site of β-TrCP1. Third, YAP might directly interact with NLRP3 and thereby mask the β-TrCP1 binding site of NLRP3.

To test the first possibility, we overexpressed YAP in iBMDMs and assessed NLRP3 serine phosphorylation. LPS stimulation led to the increase of NLRP3 serine phosphorylation, which was not affected by YAP overexpression, excluding the first possibility (Supplementary Fig. 5a). To test the second possibility, we overexpressed YAP-S366A mutant (in humans, the homologous mutant is YAP-S381A), which lost the β-TrCP1 binding capacity[31] (Supplementary Fig. 5b). However, the S366A substitution in YAP did not affected the NLRP3 and β-TrCP1 binding (Supplementary Fig. 5c), suggesting that YAP interrupts the NLRP3 and β-TrCP1 interaction independent of its direct binding to β-TrCP1. Next, we tested the third possibility and observed an association between YAP and NLRP3 (Supplementary Fig. 5d). This phenomenon was confirmed by using Proximity Ligation Assay (PLA), which could visualize the protein-protein interactions in vivo (Supplementary Fig. 5e, f). As shown by co-immunoprecipitation assay, we found that the C-terminal transactivation domain (residues 151–488) rather than the N-terminal TEAD binding domain of YAP physically interacted with NLRP3 (Supplementary Fig. 5g, h). Co-expression of wild-type YAP or its C-terminal transactivation domain but not the N-terminal TEAD-binding domain interrupted the interaction between NLRP3 and β-TrCP1 (Supplementary Fig. 5i), indicating that the NLRP3-YAP binding inhibits the NLRP3-β-TrCP1 interaction. Further, we observed that both YAP and β-TrCP1 bound to the same domain of NLRP3 (NACHT and LRR domain) (Supplementary Fig. 5j, k, l). Taken together, these results demonstrate that YAP disrupts the interaction between NLRP3 and β-TrCP1 through competing with β-TrCP1 to bind NLRP3.

**β-TrCP1 inhibits NLRP3 inflammasome activation**. To determine whether β-TrCP1 might inhibit the NLRP3 inflammasome activation, we knocked down the expression of β-TrCP1 in primary macrophages by using siRNA. Silencing of β-TrCP1 expression in primary macrophages markedly enhanced the release of IL-1β or IL-18, but did not affect the TNF-α secretion, upon stimulation by NLRP3 agonists (Fig. 6a). Accordingly, silencing of β-TrCP1 expression markedly decreased the NLRP3 expression (Fig. 6b). Moreover, upon silencing of β-TrCP1 in YAP-deficient macrophages from Yap^fl/fl lyz2-Cre mice, the decline in IL-1β or IL-18 secretion and NLRP3 expression was rescued as compared to that of in macrophages from control Yap^fl/fl mice (Fig. 6c, d), which further confirmed that YAP could suppress β-TrCP1-mediated NLRP3 degradation. These results clearly suggest that β-TrCP1 inhibits NLRP3 inflammasome activation.

**Lys380 in NLRP3 is essential for its ubiquitination and degradation**. Next, we sought to identify the β-TrCP1-mediated ubiquitination site of NLRP3. HEK293T cells were transfected with plasmids encoding a series of truncation mutants of NLRP3 (ΔPYD, ΔNACHT andΔLRR) (Supplementary Fig. 5j) and HA-ubiquitin-K27, with or without exogenous β-TrCP1 expression. Notably, deletion of the NACHT domain prevented NLRP3 ubiquitination upon β-TrCP1 co-expression (Supplementary

Fig. 6a), suggesting that β-TrCP1 mediates the ubiquitination of the NACHT domain of NLRP3. We further analyzed the most conserved lysine residues within the NACHT domain among species and constructed plasmids encoding mutated NLRP3 with individual lysine residue replaced by arginine (K/R) (Supplementary Fig. 6b). The K380R substitution rather than other point mutations abolished β-TrCP1-mediated K27-linked ubiquitination of NLRP3 (Fig. 7a). We next tested whether K27-linked ubiquitin chains on K380 might function as a degradation signal for NLRP3, and observed a significantly slower degradation rate of the K380R NLRP3 mutant as compared to that of wild-type NLRP3 (Fig. 7b). Further, co-expression of β-TrCP1 failed to promote the degradation of the K380R NLRP3 mutant (Fig. 7c).

To further address the functional importance of K380 ubiquitination, we reconstituted the NLRP3 inflammasome in HEK293T cells and observed that the expression of K380R mutant enhanced IL-1β secretion as compared to that of wild-type NLRP3 (Fig. 7d). To confirm this phenomenon in immune cells, we induced the expression of WT NLRP3 or its K380R mutant in NLRP3−/− iBMDMs using lentivirus, and then stimulated the LPS-primed iBMDMs with ATP or nigericin. We found that expression of the K380R mutant significantly increased the release of IL-1β but not TNF-α secretion as compared to that of cells expressing WT NLRP3 (Fig. 7e, f). Taken together, these data indicate that K380 of NLRP3 is an important site for the K27 ubiquitination and the degradation of NLRP3.

**YAP promotes NLRP3 inflammasome activation in vivo**. Finally, we investigated the role of YAP in NLRP3 inflammasome activation in vivo. Intraperitoneal (i.p.) injection of LPS induces IL-1β secretion and neutrophil infiltration in a NLRP3-dependent manner[35,36]. Notably, the secretion of IL-1β but not TNF was significantly reduced in Yap^fl/fl lyz2-Cre mice, as compared to that of in control Yap^fl/fl mice (Fig. 8a). Moreover, deletion of YAP in myeloid cells markedly attenuated endoxemia-induced lung injury, as evaluated by histopathology (Fig. 8b). To confirm the role of YAP in regulating NLRP3 inflammasome activation in vivo, we injected i.p. the mice with monosodium urate (MSU), which induces neutrophil infiltration and peritonitis in a NLRP3 dependent manner[35,36]. We observed that deletion of YAP in myeloid cells significantly inhibited MSU-induced IL-1β release and neutrophil infiltration as compared to that of in control Yap^fl/fl mice (Fig. 8c, d and Supplementary Fig. 7). Collectively, these findings demonstrated that YAP promotes NLRP3 inflammasome activation in vivo.

## Discussion
The Hippo pathway participates in the innate immune responses in multiple aspects, including antibacterial immune response, antiviral immune response and macrophage polarization[17,37,38]. Accumulating evidence suggest that up-regulation of YAP expression is associated with the development of inflammatory diseases, such as atherosclerosis[16], inflammatory bowel disease (IBD)[17], sepsis[18] and pancreatitis[19]. However, the underlying mechanisms by which YAP regulates inflammation and promotes the pathogenesis of inflammatory diseases are not fully understood. In this study, we demonstrated that myeloid YAP deficiency specifically impairs NLRP3 inflammasome activation both in vitro and in vivo. In addition, serum starvation- or high cell confluence-induced Hippo signaling activation that induces the degradation of YAP markedly decreased the activity of the NLRP3 inflammasome. Mechanistically, YAP maintains the stability of NLRP3 by direct binding to NLRP3, thereby blocking the accessibility of E3 ligase β-TrCP1, the latter promotes the

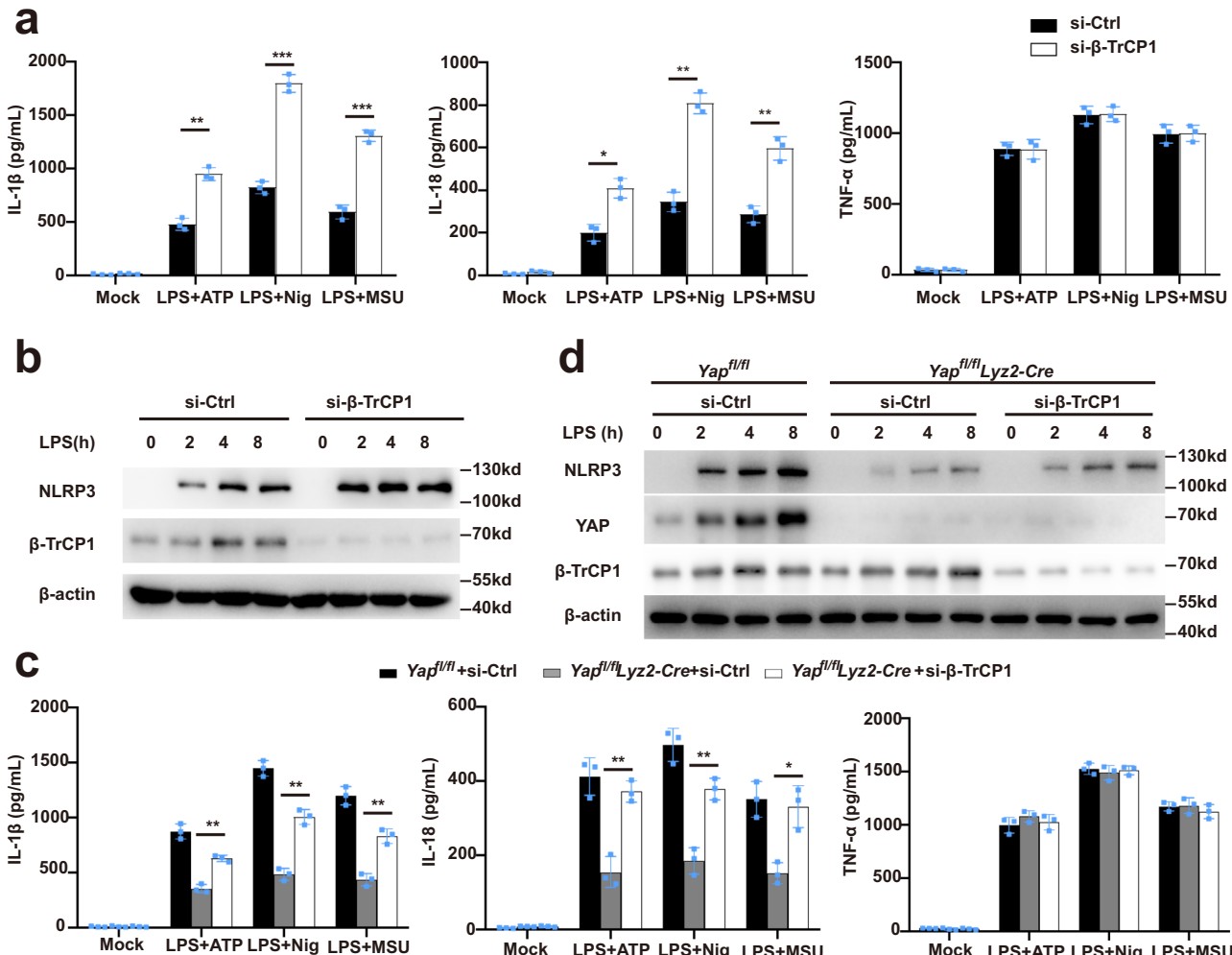

**Fig. 6 β-TrCP1 inhibits NLRP3 inflammasome activation. a** ELISA of IL-1β, IL-18, and TNF-α in supernatants from mouse peritoneal macrophages silenced of β-TrCP1, treated with indicated stimuli (mean ± SD, two-way ANOVA with Bonferroni test, β-TrCP1 siRNA vs. Ctrl siRNA, left panel: **$P = 0.0024$, ***$P = 0.0006$, ***$P = 0.0006$ in sequence; middle panel: *$P = 0.0169$, **$P = 0.0011$, **$P = 0.0081$ in sequence; $n = 3$ independent experiments). **b** Immunoblot analysis of extracts from mouse peritoneal macrophages silenced of β-TrCP1, then stimulated for indicated times with LPS. **c** ELISA of IL-1β, IL-18, and TNF-α in supernatants from $Yap^{fl/fl}$ lyz2-Cre or $Yap^{fl/fl}$ peritoneal macrophages silenced of β-TrCP1, treated with indicated stimuli (mean ± SD, two-way ANOVA with Bonferroni test, $Yap^{fl/fl}$ lyz2-Cre + β-TrCP1 siRNA vs. $Yap^{fl/fl}$ lyz2-Cre + Ctrl siRNA, left panel: **$P = 0.0023$, **$P = 0.0018$, **$P = 0.0047$ in sequence; middle panel: **$P = 0.0084$, **$P = 0.0066$, *$P = 0.0498$ in sequence; $n = 3$ independent experiments). **d** Immunoblot analysis of lysates from $Yap^{fl/fl}$ lyz2-Cre or $Yap^{fl/fl}$ mouse peritoneal macrophages silenced of β-TrCP1, treated with indicated stimuli. Similar results were obtained from three independent experiments. Source data are provided as a Source Data file.

proteasomal degradation of NLRP3 via K27-linked ubiquitination at K380 (Fig. 8e). Thus, our study provides a mechanistic insight into the regulatory roles of YAP in inflammation.

Accumulated evidence reveal that altered NLRP3 expression is associated with the pathogenesis of several inflammatory disorders, including atherosclerosis[39,40], type-2 diabetes (T2D)[41], and autoimmune diseases[42]. Our study suggests that YAP may be a potential therapeutic target to regulate NLRP3 protein expression in many inflammatory diseases. This is consistent with a recent report[43] showing elevated YAP expression in atherosclerotic plaques, which are enriched in NLRP3 expression[40,41]. It was also found that YAP protein levels correlate with plasma IL-1β levels in patients with ST-segment elevated myocardial infarction (STEMI)[43], suggesting that YAP could be a potential target for the treatment of atherosclerosis. Serum starvation and high cell confluence, the well-known cellular stresses that activate Hippo signaling, could inhibit the NLRP3 inflammasome activation in a YAP-dependent manner. Given that other cellular

stresses, including matrix stiffness, mechanical stress, and long-range hormonal signals, are able to activate the Hippo-YAP pathway, we propose that these stress signaling might also regulate the NLRP3 inflammasome activation in the lung, a highly mechanical organ[44].

Ubiquitination of NLRP3 plays an important role in the regulation of inflammasome[1,45]. To date, NLRP3 has been reported to be ubiquitinated with both K48 and K63 linkages. A series of enzymes that are responsible for K48-linked ubiquitination of NLRP3 have been identified, including PAI2[46], MARCH7[35], FBXL2[47], Trim31[48], and Cbl-b[49]. All of these bind to NLRP3 and mediate the degradation of NLRP3 either by the autophagy-lysosomal pathway or the ubiquitin-proteasome pathway. BRCC3-mediated deubiquitination of NLRP3 is required for NLRP3 inflammasome assembly and activation[50], in which the ubiquitination of NLRP3 consists of both K48 and K63 linkages. In our study, we found that NLRP3 can be ubiquitinated with K27 linkage, leading to subsequent NLRP3 degradation in the

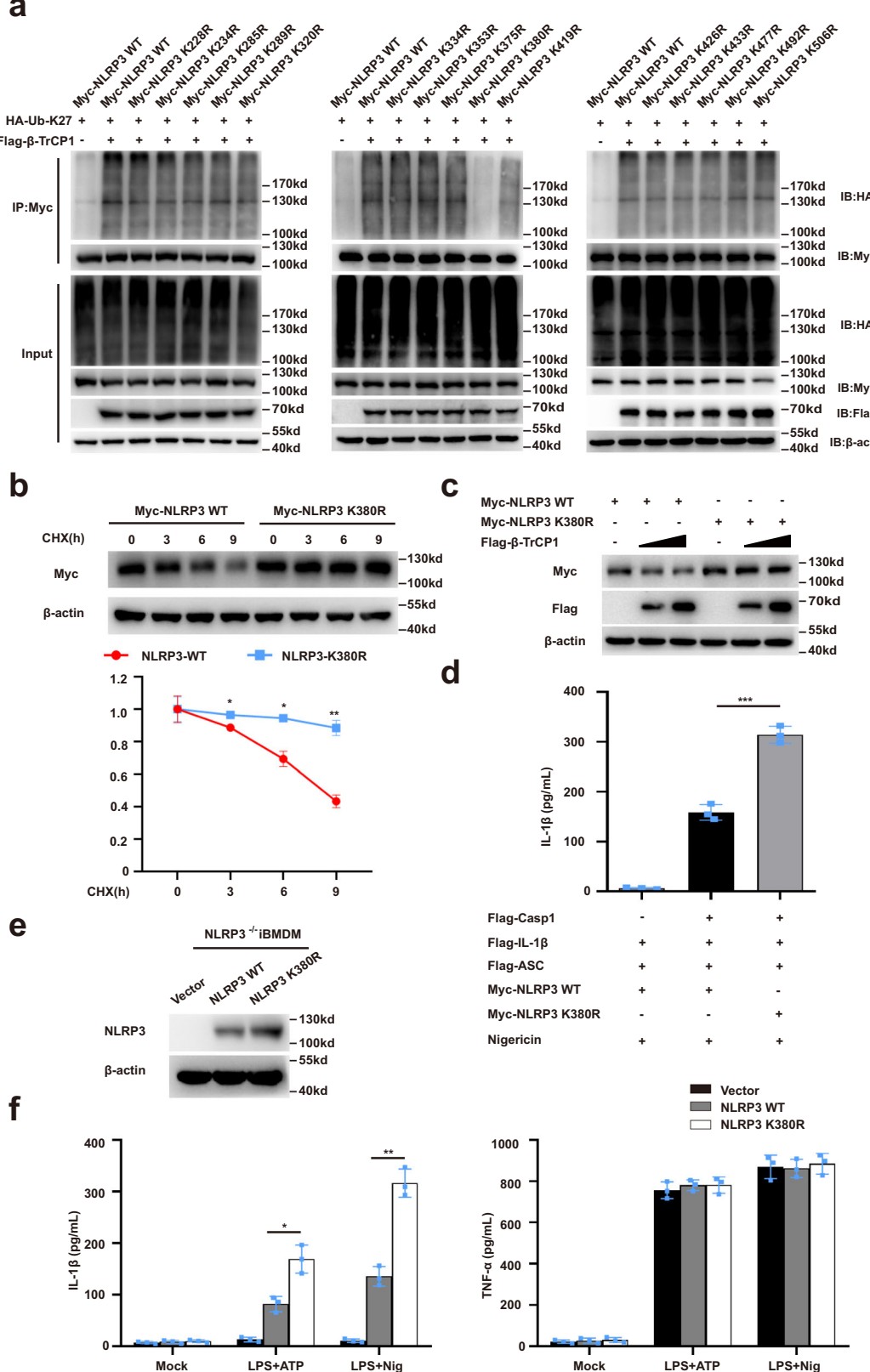

ubiquitin-proteasome pathway. This finding provides insights into the regulation of NLRP3 degradation. Although most studies have shown that K48-linked ubiquitination is the signal that mediates proteasomal degradation[51,52], our work provides compelling evidence that K27-linked ubiquitination could also lead to the proteasomal degradation of target proteins[53–55].

Moreover, we identified that Lys380 is the K27-linked ubiquitination site by screening the most-conserved lysine residues in the NACHT domain of NLRP3. When we replaced lysine with arginine, the degradation of NLRP3 was inhibited. Accordingly, cells expressing the NLRP3 K380R mutant released more IL-1β as compared to cells expressing WT NLRP3. Thus, our study

**Fig. 7 Lys380 in NLRP3 is essential for its ubiquitination and degradation. a** Immunoblot analysis of lysates from HEK293T cells transfected with HA-tagged K27-linked ubiquitin (K27-Ub), β-TrCP1 and Myc-NLRP3 or indicated mutant Myc-NLRP3, followed by IP with anti-Myc, probed with anti-HA. **b** Immunoblot analysis of lysates from HEK293T cells transfected with Myc-NLRP3, Myc-NLRP3 K380R, and then treated for various times with cycloheximide (CHX, 100 ug/mL) (top). NLRP3 and NLRP3 K380R expression levels were quantitated by measuring band intensities using "ImageJ" software. The values were normalized to actin (bottom) (mean ± SD, two-way ANOVA with Bonferroni test, NLRP3-K380R vs. NLRP3-WT lower panel: *$P = 0.0129$, *$P = 0.0321$, **$P = 0.0010$ in sequence; $n = 3$ independent experiments). **c** Immunoblot analysis of lysates from HEK293T cells transfected with Flag-β-TrCP1 and Myc-NLRP3 or Myc-NLRP3 K380R. **d** ELISA of IL-1β in supernatants from HEK293T cells transfected with ASC, pro-caspase-1, pro-IL-1β, NLRP3 or NLRP3 K380R plasmids and stimulated with 10 μM nigericin (mean ± SD, one-way ANOVA with Bonferroni test, the third group vs. second group ***$P < 0.0001$; $n = 3$ independent experiments). **e, f** NLRP3$^{-/-}$ iBMDMs were infected with lentivirus expressing mouse WT or NLRP3-K380R, respectively. Immunoblot analysis of indicated proteins (**e**), then treated these cells with LPS plus ATP or nigericin for detection of IL-1β and TNF-α in supernatants (**f**) (mean ± SD, two-way ANOVA with Bonferroni test, NLRP3-K380R vs. NLRP3-WT left panel: *$P = 0.0475$, **$P = 0.0037$ in sequence, $n = 3$ independent experiments). Similar results were obtained from three independent experiments. Source data are provided as a Source Data file.

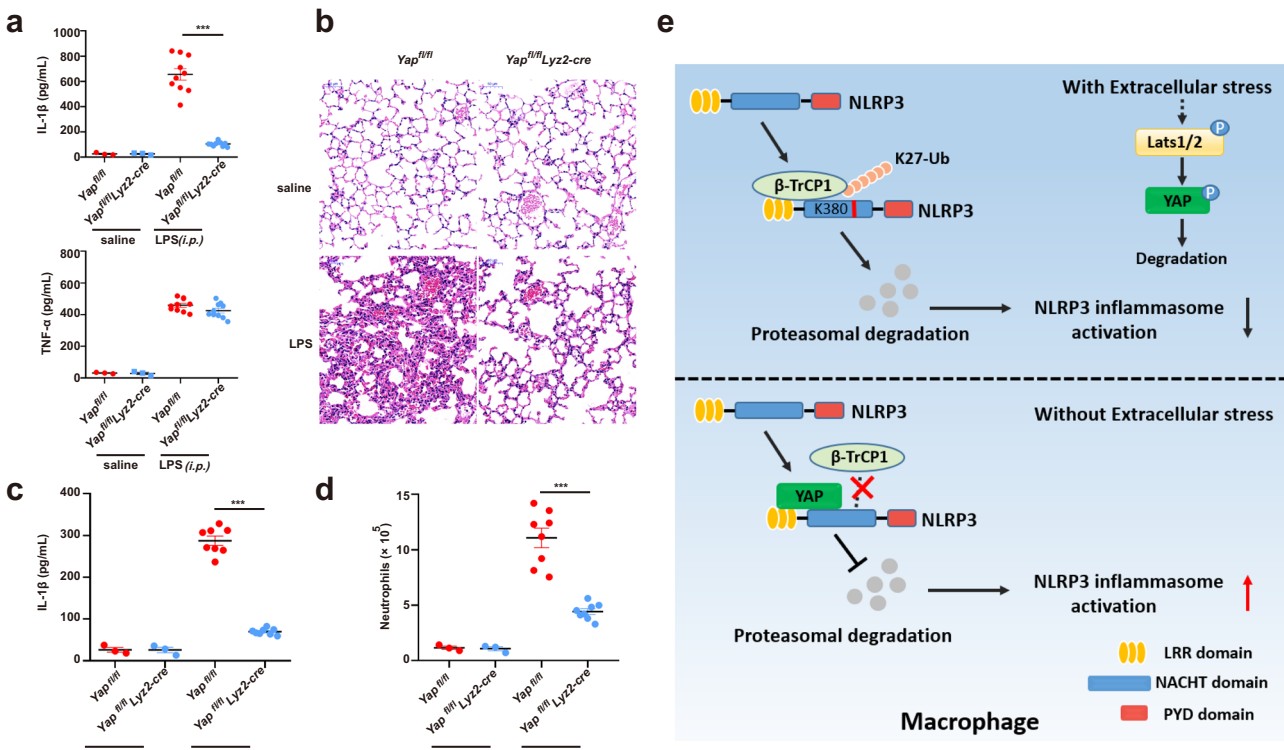

**Fig. 8 YAP promotes NLRP3 inflammasome activation in vivo. a, b** $Yap^{fl/fl}$ lyz2-Cre or $Yap^{fl/fl}$ mice were intraperitoneal injected 20 mg/kg LPS for 6 h, ELISA analysis of serum levels of IL-1β and TNF-α (**a**), H&E staining of lung tissue sections(**b**) Scale bars, 50 μm (mean ± SEM, two-way ANOVA with Bonferroni test, $Yap^{fl/fl}$ lyz2-Cre vs. $Yap^{fl/fl}$, ***$P < 0.0001$, $n = 10$ biologically independent mice). **c, d** $Yap^{fl/fl}$ lyz2-Cre or $Yap^{fl/fl}$ mice were intraperitoneally injected with 1 mg monosodium urate for 6 h, ELISA analysis of IL-1β (**c**) and quantification of neutrophils (**d**) in the peritoneal cavity fluid (mean ± SEM, two-way ANOVA with Bonferroni test, $Yap^{fl/fl}$ lyz2-Cre vs. $Yap^{fl/fl}$, ***$P < 0.0001$ (**c**), ***$P < 0.0001$ (**d**) in sequence, $n = 8$ biologically independent mice). **e** The model of YAP in NLRP3 inflammasome activation. YAP maintains the stability of NLRP3 by blocking/masking the interaction between NLRP3 and the E3 ligase β-TrCP1, which promotes the proteasomal degradation of NLRP3 via K27-linked ubiquitination at lys380. Cellular nutrient/density status, activating the Hippo-YAP pathway, impairs NLRP3 inflammasome activation in a YAP-dependent manner. Source data are provided as a Source Data file.

revealed a ubiquitination site for NLRP3 degradation and inactivation.

β-TrCP1, one of the best-characterized mammalian F-box proteins of SCF ligase, has been reported to be involved in the ubiquitination of well-established substrates, such as YAP, β-catenin, IκB, caspase-3[34,56]. All of these studies have suggested that β-TrCP1-mediated substrate ubiquitination is K48-linked. However, in current study, we did not observe that β-TrCP1-mediated K48-linked ubiquitination of NLRP3. Instead, β-TrCP1 promoted K27-linked ubiquitination of NLRP3 and its subsequent degradation. It has been well-documented that β-TrCP1's substrates contain a consensus DSG(X)$_{2+n}$S degron[33,34]. The serine residues in this degron could be phosphorylated by specific protein kinases, resulting in β-TrCP1 recognition and subsequent

ubiquitination of the substrates. We identified the 890-NSGLTS-895 motif, an "atypical" XSG(X)$_{2+N}$S motif in NLRP3, is essential for β-TrCP1 binding. However, the kinase responsible for NLRP3 serine phosphorylation remains unknown. Previous studies have shown that kinases JNK1[36], PKA[57,58], PKD[59], and AKT[60] could mediates serine phosphorylation of NLRP3. However, the reported phosphorylation site was neither serine 891 nor 895. Whether these kinases could also mediate serine 891/895 phosphorylation and whether unrecognized kinases account for NLRP3 phosphorylation, are interesting questions for future studies. Previous studies have suggested that 14-3-3 family members including 14-3-3ε, 14-3-3η, and14-3-3τ, can bind to pyrin and regulate the activation of the pyrin inflammasome[61,62]. Although we found that 14-3-3σ has no effect on the

ubiquitination of NLRP3, the role of the14-3-3 family in NLRP3 inflammasome activation deserves further investigation.

In this work, our study uncovers a role of YAP in regulation of the NLRP3 inflammasome activation, and provides potential therapeutic target to treat a number of inflammatory disorders, such as atherosclerosis, gout, colitis and sepsis.

## Methods

**Mice**. Yap^fl/fl mice and Lyz2-Cre mice on the C57BL/6 background were from Jackson laboratories[20]. Yap^fl/fl mice were crossed with Lyz2-Cre mice to obtain Yap^fl/fl lyz2-Cre mice. Wild-type C57BL/6 mice (8–10 weeks old) were purchased from Hunan SJA Laboratory Animal Co.Ltd (Changsha, China). All the animals were generated in specific pathogen-free (SPF) levels and daily cycles of 12 h of light at 23 °C and 12 h of dark at 21 °C. All animal experiments were conducted in accordance with the Institutional Animal Care and Use Committee of Central South University.

**Reagents**. Ultrapure LPS (E. coli 0111:B4), Standard LPS (E. coli 0111:B4), Pam3CSK4, ATP, Nigericin, MSU, FLA-ST, Poly(dA:dT) naked, DyLight 488-labeled secondary antibody (A120-100D2, 1:50 for immunofluorescence), Alexa Fluor 594-conjugated secondary antibody (405326, 1:50 for immunofluorescence) were purchased from InvivoGen. Lipofectamine 2000 Transfection Reagent, Pierce™ Anti-c-Myc Agarose were from ThermoFisher Scientific. Anti-Caspase-1 antibody (ab179515, 1:1000 for WB), mouse IL-18 ELISA kit (Ab218808) were purchased from Abcam. Anti-IL-1β antibody (AF-401-NA; RRID: AB_416684, 1:1000 for WB) was from RD systems. Anti-YAP1 antibody (A1002, 1:1000 for WB), Anti-TAZ antibody (A15806, 1:1000 for WB) were from ABclonal. Anti-NLRP3 antibody (Cryo-2, 1:1000 for WB, 1:400 for IP), Anti-ASC antibody (AL177, 1:1000 for WB) were purchased from Adipogen. Anti-β-actin antibody (BH10D10, 1:10000 for WB), Anti-Phospho-YAP(Ser127) antibody (4911, 1:1000 for WB), Anti-β-TrCP antibody (D13F10, 1:1000 for WB), Cell Lysis Buffer were from Cell Signaling Technology. Anti-LATS1 antibody (AF7669, 1:1000 for WB) was from Affinity.

Anti-Ub antibody (Sc-8017, 1:200 for WB), mouse immunoglobin IgG protein (SC-2025, 1:160 for IP), Protein A/G PLUS-Agarose were from Santa cruz. Anti-HA-tag (M180-3, 1:5000 for WB), Anti-DDDDK-tag (M185-3L, 1:5000 for WB), Anti-Myc-tag (M047-3, 1:5000 for WB), Anti-His-tag (D291-3, 1:5000 for WB) were from MBL. FITC anti-mouse/human CD11b (101216, 1:500 for flow cytometry) and APC anti-mouse Ly-6G (127614, 1:500 for flow cytometry) were from Biolegend. Cholera Toxin B Subunit (Choleragenoid) from Vibrio cholerae was purchased from List Biological Laboratories, INC. Anti-Flag affinity gel was from Sigma. pLenti-CRISPR v2 was from Addgene. First-Strand cDNA Synthesis SuperMix was purchased from TransGen Biotech. SYBR quantitative PCR (qPCR) Master Mix was from Vazyme Biotech. Mouse IL-1β ELISA kit (88-7013), Mouse TNF-α ELISA kit (88-7324) were purchased from eBioscience. Duolink In Situ Red Starter Kit Goat/Rabbit kit, Duolink In Situ PLA Probe Anti-Mouse MINUS were purchased from Sigma.

**Cell culture**. To obtain mouse primary peritoneal macrophages, mice were injected intraperitoneally with 3% thioglycolate. Three days later, peritoneal exudate cells were harvested and incubated. Two hours later, nonadherent cells were removed and the adherent monolayer cells were used as peritoneal macrophages. HEK293T cells were obtained from American Type Culture Collection (Manassas, VA). The cells were cultured in Dulbecco's modified Eagle's medium (DMEM) supplemented with 10% fetal bovine serum and 1% penicillin and streptomycin.

The concentration of agonists or stimuli were used as below: For NLRP3 inflammasome activation: peritoneal macrophages were primed with LPS (100 ng/mL) for 3 h followed by stimulation with 5 mM ATP (1 h); 10 μM Nigericin (1 h); 200 μg/mL MSU (6 h). For non-canonical NLRP3 inflammasome activation, macrophages were primed for 4 h with Pam3CSK4 (100 ng/mL), then treated with LPS (2 μg/mL) combined with CTB (5 μg/mL) (CTB + LPS), or LPS (2 μg/mL) transfected with FuGENE HD (0.25% v/v) for 16 h. For AIM2 inflammasome activation, Pam3CSK4- primed macrophages were transfected with Poly (dA:dT) (1 μg/mL) using Lipofectamine 2000 (3 μL/mg DNA) following the manufacturer's protocol (Invitrogen). For NLRC4 inflammasome activation, macrophages were primed with LPS (100 ng/mL) for 3 h, then transfected with Flagellin (2 μg/mL) by Lipofectamine 2000 for 1 h.

**Reconstitution of NLRP3 inflammasome in HEK293T cells**. A standard reconstitution system in HEK293T cells was referred[63]. HEK293T cells were seeded into 24-well plates at density of $2 \times 10^5$ cells per well in complete cell culture medium overnight before transfection. The cells were transfected with plasmids expressing pro-IL-1β-flag (200 ng/well), pro-caspase-1-myc (100 ng/well), ASC-myc (20 ng/well), NLRP3-myc (200 ng/well) using Linear Polyethylenimine. Twenty-four to 36 h later, replace the medium with 250 μL DMEM cell culture medium, then 10 μM nigericin was added one hour before supernatant collection.

**Plasmids and transfection**. NLRP3, caspase-1, pro-IL-1β, ASC, 14-3-3σ, and β-TrCP1 full-length sequences were obtained from mouse peritoneal macrophage cDNA and YAP full-length sequences was obtained from mouse testis tissue cDNA, then cloned into pcDNA3.1 vector that contained different tags. Deleted, truncated, and point mutants were generated by PCR-based amplification and the construct encoding the wild-type protein as the template. All constructs were confirmed by DNA sequencing. Ubiquitin and its mutants (K6, K11, K27, K29, K33, K48, K63and K27R) genes were synthesized (Sangon Biotech, Shanghai, China) and cloned into pcDNA3.1-HA eukaryotic expression vectors respectively. Plasmids were transiently transfected into HEK293T cells with Linear Poly-ethylenimine. For iBMDMs, we generate NLRP3, NLRP3 K380R, and YAP plasmids, which cloned in the lentiviral vector, pCDH-MCS-T2AcopGFP. Lentivirus was produced in HEK293T cells by co-transfection of lentiviral vectors (pCDH-MCS-EF1-copGFP), pSPAX2, and pVSV-G with a ratio of 3:2:1. Virus-containing supernatants were filtered through a 0.45-μm-pore-size filter (Millipore) and supplemented with polybrene (8 μg/mL) before adding to cells. NLRP3^−/− iBMDMs were re-transduced with lentivirus encoding NLRP3 WT or NLRP3 K380R mutant in pCDH-MCS-EF1-copGFP vectors (System Biosciences). GFP-positive cells were then sorted by flow cytometry (FACS Aria II, BD Biosciences).

**ASC oligomerization and ASC speck formation**. Peritoneal macrophages were plated in 6-well plates ($2 \times 10^6$ cells per well), then stimulated with 100 ng/mL LPS for 3 h, followed by stimulation with or without 10 μM nigericin (1 h). The cells were lysed with Triton Buffer [50 mM Tris-HCl (pH 7.5), 150 mM NaCl, 0.1% Triton X-100, 0.1 mM PMSF and EDTA-free protease inhibitor cocktail] at 4 °C for 10 min, then remove 30 μL as whole cell lysates. The rest of cell lysates were centrifuged at $6000 \times g$ at 4 °C for 15 min, the supernatant was collected and pellet was washed twice and re-suspended in 200 μL Triton Buffer, and cross-linked for 30 min at 37 °C with 2 mM disuccinimidyl suberate (DSS). The pellets were collected after centrifuged for 15 min at $6000 \times g$, and dissolved in SDS loading buffer for immunoblot analysis

For ASC speck formation, peritoneal macrophages were seeded on chamber slides and allowed to attach overnight. The next day, the cells were treated with indicated stimulators. Then the cells were fixed in 4% paraformaldehyde followed by permeabilization with 0.1% Triton X-100 for 30 min. Then the slides were blocked with phosphate-buffered saline (PBS) containing 1% Bovine serum albumin (BSA), followed by ASC and DAPI staining.

Cells were visualized by fluorescence microscope (Nikon Ti2-U).

**Proximity ligation assay**. Proximity ligation assays were carried out to detect protein-protein interactions in situ by using Duolink reagents (Sigma). Briefly, primary macrophages were cultured on a six-well object glass in RPMI medium 1640. After stimulated with LPS for 4 h or not, cells were fixation in 4% paraformaldehyde and permeabilization with PBS containing 0.1% Triton X-100, then incubated over night with primary antibody pair of different species directed to NLRP3 (1:200), YAP (1:200), β-TrCP1 (1:200). Probe incubation, ligation and amplification reaction were carried out according to the manufacturer instructions. Thus, each individual pair of proteins generated a spot that could be visualized using fluorescent microscopy (Nikon Ti2-U).

**CRISPR/Cas9-mediated generation of NLRP3^−/− iBMDM cells**. CRISPR/Cas9 genomic editing for gene deletion according to the previous publication[64]. Guide RNA sequences targeting NLRP3(5′- gtcctcctggcataccatag- 3′) with BsmBI sticky end were annealed and inserted into the lentiviral vectors pLenti-CRISPR v2 (Addgene #52961) digested with BsmBI (NEB). Lentivirus was produced in HEK293T cells by co-transfection of plenti-sgRNA plasmid, pSPAX2, and pMD2.G with a ratio of 3:2:1. Virus-containing supernatants were filtered through a 0.45-μm-pore-size filter (Millipore) and supplemented with polybrene (8 μg/mL) before adding to cells. iBMDMs then were transduced with lentivirus encoding NLRP3 guide RNA. Stable transduced cells were selected with 5 mg/mL puromycin for 72 h, and single colonies were obtained by serial dilution and amplification. Clones were identified by immunoblotting with anti-NLRP3 antibodies, and the NLRP3^−/− clone was used for the indicated analyses.

**RNA interference assay**. Peritoneal macrophages were seeded in 24 well ($2 \times 10^5$ cells per well) or 6-well plates ($1 \times 10^6$ cells per well), then transfected with siRNA using Lipofectamine RNAiMAX (Thermo Fisher Scientific) according to the manufacturer's instructions. The siRNA sequences for mouse YAP (GGUCAAA GAUACUUCUUAAUT), Mouse β-TrCP1 (CCAUCGCUGUGUGGGAUAUTT), Mouse Lats1(GCAAGUCACUCUGCUAAUUTT), Mouse Lats2(GCCUCAAUG CUGACUUGUAUT) and the negative control (UUC UCCGAACGUGUCACGU TT) were chemically synthesized by Sangon Biotech Co., Shanghai, China.

**Immunofluorescence**. Primary macrophages were stimulated with LPS for indicated hours, then cells were fixed with 4% paraformaldehyde for 15 min and permeabilized with 0.1% Triton X-100 for 10 min. After blocking with 3% BSA for 1 h, cells were incubated overnight with anti-NLRP3 antibody (1:100 in PBS containing 3% BSA) and β-TrCP1 antibody (1:100 in PBS containing 3% BSA), followed by staining with DyLight 488-labeled secondary antibody (Invitrogen) and

Alexa Fluor 594-conjugated secondary Ab (Invitrogen) (1:50 in PBS containing 3% BSA). Nuclei were co-stained with DAPI (Invitrogen). Stained cells were viewed using a confocal fluorescence microscope (SpinSR10; Olympus).

**Quantitative PCR**. Total RNA was extracted by using RNA Fast 200 kit according to the manufacturer's instructions (FASTAGEN). Complementary DNA was synthesized by using TransScript All-in-One First-Strand cDNA Synthesis Super-Mix for qPCR (TransGen Biotech) according to the manufacturer's protocols. Quantitative PCR was performed using SYBR Green (Vazyme Biotech) on a LightCycler 480 (Roche Diagnostics), and data were normalized by the level of β-actin expression in each individual sample. The $2^{-\Delta\Delta CT}$ method was used to calculate relative expression changes.

**Immunoprecipitation and western blot**. HEK293T cells after transfection or peritoneal macrophages after stimulation, then lysed in IP buffer, which contained 1% (v/v) Nonidet P-40, 50 mM Tris-HCl (pH 7.4), 50 mM EDTA, 150 mM NaCl, and a protease inhibitor cocktail, pre-cleared cell lysates were then subjected to anti-Flag M2 affinity gel or anti-Myc affinity gel for 2 h or with specific Abs and protein G plus-agarose overnight, then washed four times with IP buffer. Immunoprecipitates were eluted by boiling with 1% (w/v) SDS sample buffer.

For ubiquitination analysis, cells were lysed in IP buffer supplemented with 10 mM NEM (Sigma), 1% SDS, protease inhibitor cocktail and PMSF. Obtained cell lysates were subsequently boiled for 5 min at 95 °C, diluted to 0.1% SDS with IP buffer, and immunoprecipitated.

For immunoblot analysis, cells were lysed with CLB buffer (CST) supplemented with protease inhibitor cocktail and PMSF, and then protein concentrations in the extracts were measured with a bicinchoninic acid assay (Pierce). Equal amounts of extracts were separated by sodium dodecyl sulphate–polyacrylamide gel electrophoresis, and then they were transferred onto nitrocellulose membranes for immunoblot analysis.

**ELISA assay for cytokines**. Levels of IL-1β, TNF-α, and IL-18 collected from cell culture and sera were determined using quantitative ELISA kits (eBioscience or abcam) according to the manufacturer's instructions.

**In vivo LPS challenge**. *Yap^{fl/fl} lyz2-Cre* mice and *Yap^{fl/fl}* littermates were injected intraperitoneally with LPS (20 mg/kg body weight), after 6 h, the mice were killed, and the serum concentrations of IL-1β, and TNF-α were measured by ELISA.

**MSU-induced peritonitis in vivo**. *Yap^{fl/fl} lyz2-Cre* mice and *Yap^{fl/fl}* littermates were i.p. injected with 1 mg MSU (dissolved in 500 μL PBS) for 6 h. Peritoneal cavities were washed with 10 mL ice-cold PBS. The peritoneal lavage fluids were collected and concentrated for ELISA analysis with Amicon Ultra 10 K filter (UFC900308) from Millipore. Peritoneal exudate cells were collected and analyzed by FACS.

**Statistical analysis**. The data were analyzed by GraphPad Prism 8.0 software and were presented as the mean ± SD or mean ± SEM. Two-tailed unpaired Student's *t*-test was used to compare the differences between two groups. One-way ANOVA or two-way ANOVA with Bonferroni's post hoc test were used for comparisons of more than two groups. Differences were considered significant when $*P < 0.05$, $**P < 0.01$ and $***P < 0.001$.

**Reporting summary**. Further information on research design is available in the Nature Research Reporting Summary linked to this article.

## Data availability
The authors declare that the data supporting the findings of this study are available within the paper and its supplementary information files or available from the corresponding author upon reasonable request. Source data are provided with this paper.

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

## Acknowledgements

We thank Dr. Feng Shao (National Institute of Biological Sciences) for providing immortalized mouse macrophages. We thank Qianqian Xue for assisting in raising the animals. This work was supported by National Natural Science Foundation of China (81801967, 82025021, 81930059, 81971893), Innovation-driven Project of Central South University (2018CX030, 2019CX013), Natural Science Foundation of Hunan Province, China (2020JJ5873), Fundamental Research Funds for the Central Universities of Central South University (CX20190076).

## Author contributions

K.Z. and B.L. supervised the whole project. K.Z. and B.L. acquired funding for the study. K.Z. designed the research. K.Z., D.W., and B.L. wrote the manuscript. D.W. and Y. Z. analyzed the data. D.W., Y.Z., X.X., Y.P., J.L., R.L., L.H., L.L., and N.Z. performed the experiments; J.W. raised the animals and helped with data analyses and discussions; S.Y. constructed NLRP3$^{-/-}$ iBMDMs.

## Competing interests

The authors declare no competing interests.
