## [Peer Review File · Nature Communications]

REVIEWER COMMENTS>

Reviewer #1 (Remarks to the Author):

The authors are exploring the role of YAP in the activation and maintenance of the inflammasome. They use two models to test this. HEK293 cells in which YAP is either overexpressed or knocked down or peritoneal macrophages (PM) isolated from mice with a myeloid specific YAP KO. First they show that YAP knockdown in HEK293 or KO in PM reduces inflammasome activation in response to several stimuli. Next they show that YAP inhibits the proteasomal degradation of NLRP3 and that this inhibition is much stronger when YAP is predominantly cytoplasmic compared to when it is mostly nuclear. They show that YAP reduces K27-linked ubiquitination of NLRP3. The beta-TrCP ligase is required for K27-linked ubiquitination of NLRP3 and YAP expression reduces beta-TrCP binding to and ubiquitination of NLRP3. In contrast, YAP loss increases it. They identify Lys380 in NLRP3 as essential for this ubiquitination and degradation. Lastly, they show that YAP-mediated prevention of NLRP3 degradation promotes inflammasome activation in vitro and in a mouse model in which they treat mice with LPS.

Despite several grammatical errors, the paper is well and the conclusions are clear. The data are strong, well controlled, and for the most part support their main conclusions. The work is novel as it is the first to link YAP directly to inflammasome activation, and it would be of interest to readers. However there are some major concerns that need to be addressed:

1. While the authors clearly define a role for YAP in the activation of the inflammasome using the KO mouse model and in vitro cell models, I have 3 related comments about the significance and biological relevance of the pathway they have described:
 - a. They only study the effects of overexpression of exogenous YAP or complete loss of endogenous YAP, but it is not clear when either of these situations would occur during either normal or pathological inflammatory processes. Is there a biologically relevant context where YAP levels or localization are altered in macrophages and do those changes impact the inflammasome? Can they show that other stimuli that impact YAP expression or localization influence the inflammasome in a YAP-dependent manner? Without such experiments it is unclear how critical this pathway would be outside the specific context they have studied here.
 - b. The majority of their experiments use in vitro assays in which they artificially induce inflammasome activation and the only in vivo experiment is an LPS treatment of mice. Thus, it remains possible that this pathway is only important in the artificial context of the models they have used. Is there a more biologically relevant model of inflammation where YAP is also required for inflammasome activation?
 - c. They do not include any human relevance. Are YAP levels altered in human diseases with dysregulated inflammation and can this be correlated to NLRP3 protein expression. When is this pathway important and under what circumstances is it a therapeutic target as they suggest?
2. The authors show that YAP regulates NLRP3 by preventing its degradation by beta-TrCP, but it remains unclear how. They imply this is by YAP binding to beta-TrCP and presumably preventing beta-TrCP from binding NLRP3, but this potential mechanism was not tested experimentally. Mutant forms of YAP unable to bind and/or be ubiquitinated by beta-TrCP exist and could easily be used in the HEK293 system to test this. This question is important because it is puzzling how this mechanism would work. Beta-TrCP binding to YAP is a major regulatory mechanism of YAP protein levels. When beta-TrCP binds YAP it triggers YAP's ubiquitination and degradation, which would reduce YAP protein expression, and thus likely allow for beta-TrCP mediated degradation of NLRP3. So how is

YAP-mediated stabilization NLRP3 sustained when YAP binding to beta-TrCP would likely lead to YAP degradation? The fact that this proposed mechanism is largely explored in cells overexpressing exogenous YAP raises the concern that this mechanism only exists when YAP levels are kept artificially high and cytoplasmic, which may not be biologically relevant. The authors should confirm that YAP binding to beta-TrCP is the mechanism and then at least discuss how this would be sustained. One would predict that LATS-mediated phosphorylation of YAP would promote NLRP3 stabilization because LATS phosphorylation of YAP triggers beta-TrCP binding. They could therefore test if LATS1 and LATS2 knockdown by siRNA promotes stabilization of NLRP3 by endogenous YAP in their wildtype macrophages as this would alleviate the concern they only used overexpression models to link YAP/ beta-TrCP to NLRP3 stabilization.

3. The data provided in Figure 2 are not sufficient to show that YAP inhibits NLRP3 independent of its transcriptional activity. While the authors show that overexpressed murine YAP remains mostly cytoplasmic while the YAP S112A mutant is largely nuclear and unable to prevent NLRP3 degradation, this does not prove that YAP transcriptional activity is dispensable for NLRP3 stabilization. Does a form of YAP lacking its transactivation domain or unable to bind TEADs still prevent NLRP3 degradation? Furthermore, they do not confirm that the YAP S112A they generated is transcriptionally active as one would predict? Fractionation experiments in macrophages are also not sufficient as some nuclear YAP would be expected, but perhaps not detected in the nuclear fraction. I am not sure its necessary to answer these questions as they provide a mechanism and whether this requires transcriptional activity does not drastically alter their overall conclusions. However, if they are going to claim its independent of transcriptional activity they need to prove it. Otherwise its seems appropriate to conclude that NLRP3 stabilization is stronger when YAP is cytoplasmic than when its nuclear, which may suggest this mechanism is independent of YAP's transcriptional activity.

- The authors do not test a role for TAZ. Is TAZ expressed in macrophages? If so one would predict it would also regulate NLRP3 since TAZ has the same Beta-TrCP binding motif as YAP as well as another Beta-TrCP binding site in its N-terminus. So if it is expressed why would it not at least partially compensate for YAP loss? I am not suggesting they do several experiments to address this, but if macrophages express TAZ it is worth testing if it can also promote NLRP3 stabilization and then discussing why its not sufficient in the absence of YAP.

Minor:

- The statistical tests used to establish significance are not appropriate in several places. They indicate they use T tests to compare two groups, but this is not appropriate for experiments that contain more than 2 groups unless the authors have corrected for multiple comparisons, and there is no indication that this is the case. With more than 2 groups ANOVA is a better test and should be used. If ANOVA is not possible, then the authors would need to correct for multiple comparisons.

- The order of the panels in Figure 1 is confusing DE appear before C.

- I appreciate that there are word limits, but the first section is hard to follow as the authors do not explain the rationale for the various proteins they blot for. A reader must understand the inflammasome well to interpret this data (for example most readers will not know what ASC nucleation-induced oligomerization and ASC speck formation are and why they are important . Similarly, is the fact that "The activation of AIM2 and NLRC4 inflammasomes by poly (dA:dT) and FLA transfection, respectively, were not affected (Fig.1a)" important and/or expected. Again only an expert on the inflammasome will know how to interpret this. A few short sentences to explain the reason for key experiments will help the reader interpret the data.

Reviewer #2 (Remarks to the Author):

The anonymous author(s) found that YAP specifically promoted NLRP3 inflammasome activation. Interestingly, YAP inhibited the proteasomal degradation of NLRP3 independent of its transcriptional activity. YAP somehow disrupted the interaction between NLRP3 and beta-TrCP, the latter of which was shown to interact with YAP. Whereas a series of enzymes that are responsible for K48-linked ubiquitination of NLRP3 have been identified, beta-TrCP-mediated polyubiquitylation seems to be formed by K27-linked linkages. This paper clearly showed that beta-TrCP promotes degradation of NLRP3, thereby inhibiting NLRP3 inflammasome activation in vitro and in vivo, which is blocked by YAP.

In this paper, the author(s) provide evidence that YAP in the cytoplasm acts as an activator for the inflammasome. This is a new finding in this area, an interesting subject, and an important biological and medical issue. Basically, the data is clean and consistent. In addition to cytological experiments, verification experiments have been conducted on individual mice, and I think the overall quality of the paper is quite high. It is also true, however, that some mechanistic insights are somewhat lacking. If improved, the data presented in this paper could be a very important discovery.

Major comments:

1) The major flaw of this paper is the lack of mechanistic insights for how YAP inhibits the binding of NLRP3 and beta-TrCP. The author(s) claim that YAP binds to beta-TrCP and inhibits its binding to NLRP3, but no data directly support this hypothesis. Does YAP attach to the binding surface of NLRP3 and beta-TrCP to prevent them from binding? The most convincing way to verify this is to generate these three recombinant proteins in vitro and perform binding experiments in various combinations.

2) In general, the binding of beta-TrCP to its substrate requires phosphorylation of serines at the degron sequence (DSGXXS) present in the substrate protein. Does NLRP3 have such degron sequences? What is the kinase responsible for the phosphorylation of serines in the degron? YAP may inhibit this phosphorylation pathway at some point to stabilize NLRP3. This possibility must be examined.

3) Fig. 4d and 4e are very important data when considering the mechanism of the proinflammatory function by YAP, but it is a delicate data with small and subtle differences, which reduces the reliability of the paper. In particular, given that the immunoblot analysis of Fig. 4e showing the abundance of immunoprecipitated NLRP3 appears to have been performed outside of the range in which quantification is clearly maintained, a small increase in the abundance of beta-TrCP bound to NLRP3 cannot be justified by a single experiment. For both experiments, multiple experiments should be performed to adequately quantify the bands using statistical validation.

4) YAP is regulated in various ways by signals from upstream kinases (LATS, etc.). In this paper, such upstream involvement is completely ignored, but what happens to this proinflammatory function of YAP when LATS is activated or silenced?

Minor comments:

1) The authors are completely mistaken about the use of the term "SCF/beta-TrCP." SCF/beta-TrCP refers to the entire complex of four proteins: Skp1, Cul1, Rbx1, and beta-TrCP, and descriptions such

as “FLAG-SCF/beta-TrCP” are incorrect. The readers do not understand which of these four proteins is attached with FLAG tag. From the context, it is likely that the authors have attached a FLAG tag to beta-TrCP, so it should be described as “FLAG-beta-TrCP.”

2) There are two paralogs in beta-TrCP, beta-TrCP1 (Fbxw1) and beta-TrCP2 (Fbxw 11). Given that the phenotypes of the knockout mice are completely different from each other, their biological functions are thought to be very different. In this article, the molecular name is simply described as “beta-TrCP,” and there is no precise description of whether beta-TrCP1 or beta-TrCP2 was used in the experiments. Unless this point is properly described, it is not possible to conduct follow-up experiments that can be verified by a third party, and this should be improved.

3) Figure 1e: What antibody was used in the upper blot (pellet)?

Reviewer #3 (Remarks to the Author):

This manuscript describes the regulation of NLRP3 by K27 ubiquitination by the E3 ligase, SCF β -TrCP. Yap maintains stability of NLRP3 by blocking its interaction with the E3 ligase and preventing K27 ubiquitination and degradation of NLRP3. Overall, the findings are clear and conclusions are well supported by the data. I had some minor comments.

Minor comments:

Spelling errors and grammatical errors throughout – should be carefully screened.

1. Figure 3c

Authors transfect a range of ubiquitin mutants into cells and examine the ubiquitination of NLRP3. Only the K27 mutant appears to ubiquitinate NLRP3. This is surprising as the authors state that NLRP3 can undergo K48 and K63 ubiquitination, yet experimentally, they do not appear to have validated this. The authors should comment on this result. The authors convincingly show that YAP reduces K27 ubiquitination. Furthermore, the authors conclude that YAP does not prevent ubiquitination at any other lysine residue. This claim is invalid as transfection of HEK cells with any other ubiquitination mutant did not result in an increase in ubiquitination.

2. Figures 3a, 3c (lane 1) and 4a

The HA input IB shows banding in all lanes at 130 kDa even those not designated as containing transfected HA tagged proteins. I was unable to determine whether the authors intended to indicate that the HA tag was an alternative molecular weight. If this is the case, the correct HA band needs to be made clearer. Alternatively, this may be a labelling error and this needs to be corrected. Otherwise this brings into question the validity of all these IPs.

3. Overall mechanism:

Authors state that YAP interacts with SCF β -TrCP based on previous references (Beginning of results section titled Yap suppresses E3 ligase SCF β -TrCP). And from their diagram (Figure 7) the authors indicate a direct interaction of YAP with SCF β -TrCP, however they have not showed any data demonstrating a direct interaction between these two proteins. An alternative to this is binding to 14-3-3 – a known interacting partner for Yap, which has also been shown to bind to pyrin (Jeru et al *ARTHRITIS & RHEUMATISM* Vol. 52, No. 6, June 2005, pp 1848–1857). This is something should be commented on.

Reviewer #1 (Remarks to the Author):

The authors are exploring the role of YAP in the activation and maintenance of the inflammasome. They use two models to test this. HEK293 cells in which YAP is either overexpressed or knocked down or peritoneal macrophages (PM) isolated from mice with a myeloid specific YAP KO. First they show that YAP knockdown in HEK293 or KO in PM reduces inflammasome activation in response to several stimuli. Next they show that YAP inhibits the proteasomal degradation of NLRP3 and that this inhibition is much stronger when YAP is predominantly cytoplasmic compared to when it is mostly nuclear. They show that YAP reduces K27-linked ubiquitination of NLRP3. The beta-TrCP ligase is required for K27-linked ubiquitination of NLRP3 and YAP expression reduces beta-TrCP binding to and ubiquitination of NLRP3. In contrast, YAP loss increases it. They identify Lys380 in NLRP3 as essential for this ubiquitination and degradation. Lastly, they show that YAP-mediated prevention of NLRP3 degradation promotes inflammasome activation in vitro and in a mouse model in which they treat mice with LPS.

Despite several grammatical errors, the paper is well and the conclusions are clear. The data are strong, well controlled, and for the most part support their main conclusions. The work is novel as it is the first to link YAP directly to inflammasome activation, and it would be of interest to readers. However there are some major concerns that need to be addressed:

1. While the authors clearly define a role for YAP in the activation of the inflammasome using the KO mouse model and in vitro cell models, I have 3 related comments about the significance and biological relevance of the pathway they have described:

- a. They only study the effects of overexpression of exogenous YAP or complete loss of endogenous YAP, but it is not clear when either of these situations would occur during either normal or pathological inflammatory processes. Is there a biologically relevant context where YAP levels or localization are altered in macrophages and do those changes impact the inflammasome? Can they show that other stimuli that impact YAP

expression or localization influence the inflammasome in a YAP-dependent manner? Without such experiments it's unclear how critical this pathway would be outside the specific context they have studied here.

b. The majority of their experiments use *in vitro* assays in which they artificially induce inflammasome activation and the only *in vivo* experiment is an LPS treatment of mice. Thus, it remains possible that this pathway is only important in the artificial context of the models they have used. Is there a more biologically relevant model of inflammation where YAP is also required for inflammasome activation?

c. They do not include any human relevance. Are YAP levels altered in human diseases with dysregulated inflammation and can this be correlated to NLRP3 protein expression. When is this pathway important and under what circumstances is it a therapeutic target as they suggest?

2. The authors show that YAP regulates NLRP3 by preventing its degradation by beta-TrCP, but it remains unclear how. They imply this is by YAP binding to beta-TrCP and presumably preventing beta-TrCP from binding NLRP3, but this potential mechanism was not tested experimentally. Mutant forms of YAP unable to bind and/or be ubiquitinated by beta-TrCP exist and could easily be used in the HEK293 system to test this. This question is important because it is puzzling how this mechanism would work. Beta-TrCP binding to YAP is a major regulatory mechanism of YAP protein levels. When beta-TrCP binds YAP it triggers YAP's ubiquitination and degradation, which would reduce YAP protein expression, and thus likely allow for beta-TrCP mediated degradation of NLRP3. So how is YAP-mediated stabilization of NLRP3 sustained when YAP binding to beta-TrCP would likely lead to YAP degradation? The fact that this proposed mechanism is largely explored in cells overexpressing exogenous YAP raises the concern that this mechanism only exists when YAP levels are kept artificially high and cytoplasmic, which may not be biologically relevant. The authors should confirm that YAP binding to beta-TrCP is the mechanism and then at least discuss how this would be sustained. One would predict that LATS-mediated phosphorylation of YAP would promote NLRP3 stabilization because LATS phosphorylation of YAP triggers beta-TrCP binding. They could therefore test if LATS1 and LATS2 knockdown by siRNA promotes stabilization of NLRP3 by endogenous YAP in their wildtype

macrophages as this would alleviate the concern they only used overexpression models to link YAP/ beta-TrCP to NLRP3 stabilization.

3. The data provided in Figure 2 are not sufficient to show that YAP inhibits NLRP3 independent of its transcriptional activity. While the authors show that overexpressed murine YAP remains mostly cytoplasmic while the YAP S112A mutant is largely nuclear and unable to prevent NLRP3 degradation, this does not prove that YAP transcriptional activity is dispensable for NLRP3 stabilization. Does a form of YAP lacking its transactivation domain or unable to bind TEADs still prevent NLRP3 degradation? Furthermore, they do not confirm that the YAPS112A they generated is transcriptionally active as one would predict? Fractionation experiments in macrophages are also not sufficient as some nuclear YAP would be expected, but perhaps not detected in the nuclear fraction. I am not sure its necessary to answer these questions as they provide a mechanism and whether this requires transcriptional activity does not drastically alter their overall conclusions. However, if they are going to claim its independent of transcriptional activity they need to prove it. Otherwise its seems appropriate to conclude that NLRP3 stabilization is stronger when YAP is cytoplasmic than when its nuclear, which may suggest this mechanism is independent of YAP's transcriptional activity.

- The authors do not test a role for TAZ. Is TAZ expressed in macrophages? If so one would predict it would also regulate NLRP3 since TAZ has the same Beta-TrCP binding motif as YAP as well as another Beta-TrCP binding site in its N-terminus. So if it is expressed why would it not at least partially compensate for YAP loss? I am not suggesting they do several experiments to address this, but if macrophages express TAZ it is worth testing if it can also promote NLRP3 stabilization and then discussing why its not sufficient in the absence of YAP.

Minor:

- The statistical tests used to establish significance are not appropriate in several places. They indicate they use T tests to compare two groups, but this is not appropriate for experiments that contain more than 2 groups unless the authors have corrected for multiple comparisons, and there is no indication that this is the case. With more than 2

groups ANOVA is a better test and should be used. If ANOVA is not possible, then the authors would need to correct for multiple comparisons.

- The order of the panels in Figure 1 is confusing DE appear before C.

- I appreciate that there are word limits, but the first section is hard to follow as the authors do not explain the rationale for the various proteins they blot for. A reader must understand the inflammasome well to interpret this data (for example most readers will not know what ASC nucleation-induced oligomerization and ASC speck formation are and why they are important. Similarly, is the fact that “The activation of AIM2 and NLRC4 inflammasomes by poly (dA:dT) and FLA transfection, respectively, were not affected (Fig.1a)” important and/or expected. Again only an expert on the inflammasome will know how to interpret this. A few short sentences to explain the reason for key experiments will help the reader interpret the data.

Responses to comments

According to the insightful comments, we carefully revised the manuscript and performed several additional experiments. By doing so, we have strengthened mechanistic details and the physiological relevance of our findings. The point-by-point answers to the comments and suggestions were listed as below.

Major comments

1a. According to the comments, we determined whether cellular stresses that activate Hippo signaling could affect the activation of NLRP3 inflammasome.

Cells were subjected to serum starvation or high cell confluence, both of which are well-known cellular stress conditions that activate Hippo signaling (Genes Dev. 2012, 26, 2138-2143; Cell, 2012, 150, 780-791). Serum starvation or high cell confluence significantly inhibited the NLRP3 inflammasome-dependent IL-1 β or IL-18 release but not the TNF- α secretion in wild-type (WT) but not YAP deficient macrophages (**Fig.3a, b, Supplementary Fig.3a, b in the revised version**). Further, silencing of the Lats1/2 expression in macrophages prevented the decrease of YAP protein expression, and reversed the suppression of NLRP3 inflammasome activation during serum starvation (**Fig.3c, Supplementary Fig.3c, d in the revised version**). Thus, these results indicate that cellular nutrient/ density status suppresses the NLRP3 inflammasome activation by activating the Hippo-YAP pathway.

1b. In addition to the endotoxemia model, we adopted a sterile peritonitis model by i.p. injection of monosodium urate (MSU), which activates the NLRP3 inflammasome (Nature. 2006, 440,237-241; Cell. 2015,160, 62-73). We found that selective deletion of *Yap* in myeloid cells (*Yap^{fl/fl} lyz2-cre⁺*) markedly inhibited MSU-induced IL-1 β release and the neutrophil infiltration (**Fig.8c, d in the revised version**).

1c. We appreciate this valuable comment. The upregulated expression of NLRP3 is associated with the pathogenesis of several inflammatory disorders, including

atherosclerosis (Atherosclerosis.2017,267:127-138; Circ Res. 2020, 126(9):1260-1280), type-2 diabetes (T2D) (Diabetes 2013, 62(1): 194-204), and autoimmune diseases (Biomedecine & pharmacotherapie 2020, 130: 110542). Our study suggests that YAP might be a potential therapeutic target for the regulation of NLRP3 protein expression. This is in line with the findings that the expression of both YAP and NLRP3 is elevated in atherosclerotic plaques (Cell Rep. 2020, 4;32(5):107990; Atherosclerosis.2017,267:127-138; Circ Res. 2020, 126(9):1260-1280). It was also found that YAP protein levels correlate with plasma IL-1 β concentrations in patients with ST-segment elevated myocardial infarction (STEMI) (Cell Rep. 2020, 4;32(5):107990). In current study, serum starvation and high cell confluence, well-known cellular stresses that activate the Hippo signaling, could significantly inhibit the NLRP3 inflammasome activation in a YAP-dependent manner. Given that cellular stresses, such as matrix stiffness and mechanical stress could also activate the Hippo-YAP pathway, we propose that YAP might be critical for the regulation of NLRP3 expression in the lung, a highly mechanical organ (Nature 2019, 573(7772): 69-74). We added the discussion to the revised manuscript.

2. This is a very important question. We appreciate the valuable suggestions.

Three possible explanations may explain the underlying mechanisms by which YAP inhibits the binding of NLRP3 and β -TrCP1. (1) YAP might affect the serine phosphorylation of NLRP3, and thereby impairing the capacity of NLRP3 to bind β -TrCP1 (FEBS J, 2020 Oct 6; Oncogene,2004, 23(11): 2028-2036). (2)As suggested in the manuscript, YAP could interact with β -TrCP1 and thereby masking the NLRP3 binding site in β -TrCP1. (3) YAP could interact with NLRP3 and thereby masking the β -TrCP1 binding site in NLRP3. To test the first possibility, we overexpressed YAP in iBMDMs, and assessed serine phosphorylation in immunoprecipitated NLRP3 (Mol Cell, 2017 Oct 5;68(1):185-197). We observed increased phosphorylation of serine on NLRP3 upon LPS stimulation; however, YAP overexpression had no effect on the serine phosphorylation of NLRP3, excluding the first possibility (**Supplementary Fig.5a in the revised version**). To test the second possibility, cells were transfected

with plasmids expressing the YAP-S366A mutant (in humans, the homologous mutant is YAP-S381A), which lost the capacity to bind β -TrCP1 (Genes Dev,2010,24(1):72-85). However, the binding between NLRP3 and β -TrCP1 was not affected by the S366A substitution (**Supplementary Fig.5b, c in the revised version**), suggesting that YAP interrupt the interaction between NLRP3 and β -TrCP1 not through the direct binding to β -TrCP1. Next, we explored the third possibility and observed an interaction between YAP and NLRP3 *in vitro* and *in vivo* (**Supplementary Fig.5d, e, f in the revised version**). As shown by co-immunoprecipitation, the C-terminal transactivation domain (residues 151-488) of YAP interacted with NLRP3, while the N-terminal TEAD binding domain of YAP (residues 1-150) did not (**Supplementary Fig.5g, h in the revised version**). In contrast to wild-type YAP or its C-terminal transactivation domain, the N-terminal TEAD-binding domain failed to interrupt the interaction between NLRP3 and β -TrCP1(**Supplementary Fig.5i in the revised version**), suggesting that the YAP-NLRP3 binding blocks the association between β -TrCP1 and NLRP3. Finally, we observed both YAP and β -TrCP1 binding to the same domain of NLRP3 (NACHT and LRR domain) (**Supplementary Fig.5j, k, l in the revised version**). Together, these data demonstrated that YAP competes with β -TrCP1 to bind NLRP3.

Although β -TrCP1 can target YAP for degradation, we and others (Cell Rep. 2020 32(5):107990) observed that LPS triggers elevated expression of YAP in macrophages (**Fig. 2a in the revised version**). Thus, we reasoned that YAP might have a role in regulating the expression of NLRP3.

As suggested, we checked the function of Last1/2 in the NLRP3 inflammasome activation. Silencing Lats1/2 expression prevented the decrease in YAP protein (**Supplementary Fig.3c, d in the revised version**) and the inhibition of NLRP3 inflammasome activation upon serum starvation (**Fig.3c in the revised version**).

3. We appreciate the valuable suggestions. As pointed out by referee, our results could not exclude the possibility that YAP transcriptional activity is dispensable for NLRP3 stabilization. Actually, we observed that C-terminal transactivation domain (residues 151-488) of YAP blocked the association between NLRP3 and β -TrCP1

(Supplementary Fig.5i in the revised version). It is very likely that the protein-protein interaction contributes to the regulatory role of YAP in NLRP3 stability. Thus, we did not further explore whether YAP inhibits NLRP3 independent of its transcriptional activity. As suggested, we revised the conclusion that cytoplasmic YAP plays a dominant role in maintaining the stability of NLRP3 as compared to nuclear YAP.

In contrast to lung tissues, macrophages did not express detectable TAZ in the presence or absence of LPS stimulation (**Supplementary Fig.3e in the revised version**). Thus, we did not further explore the role of TAZ in the NLRP3 inflammasome activation.

Minor

We appreciated the valuable suggestion and corrected the statistical tests. For comparison of two groups, Two-tailed unpaired Student's t test was used. For comparison of more than 2 groups, One-Way ANOVA or Two-Way ANOVA with Bonferroni's post hoc test were used.

According to the comments, we corrected the order of the panels in Figure 1, and added more background information of inflammasome to the revised manuscript, so readers can better understand the purpose of our experiments.

Reviewer #2 (Remarks to the Author):

The anonymous author(s) found that YAP specifically promoted NLRP3 inflammasome activation. Interestingly, YAP inhibited the proteasomal degradation of NLRP3 independent of its transcriptional activity. YAP somehow disrupted the interaction between NLRP3 and beta-TrCP, the latter of which was shown to interact with YAP. Whereas a series of enzymes that are responsible for K48-linked ubiquitination of NLRP3 have been identified, beta-TrCP-mediated polyubiquitylation seems to be formed by K27-linked linkages. This paper clearly showed that beta-TrCP promotes degradation of NLRP3, thereby inhibiting NLRP3 inflammasome activation in vitro and in vivo, which is blocked by YAP.

In this paper, the author(s) provide evidence that YAP in the cytoplasm acts as an activator for the inflammasome. This is a new finding in this area, an interesting subject, and an important biological and medical issue. Basically, the data is clean and consistent. In addition to cytological experiments, verification experiments have been conducted on individual mice, and I think the overall quality of the paper is quite high. It is also true, however, that some mechanistic insights are somewhat lacking. If improved, the data presented in this paper could be a very important discovery.

Major comments:

1) The major flaw of this paper is the lack of mechanistic insights for how YAP inhibits the binding of NLRP3 and beta-TrCP. The author(s) claim that YAP binds to b-TrCP and inhibits its binding to NLRP3, but no data directly support this hypothesis. Does YAP attach to the binding surface of NLRP3 and beta-TrCP to prevent them from binding? The most convincing way to verify this is to generate these three recombinant proteins in vitro and perform binding experiments in various combinations.

2) In general, the binding of beta-TrCP to its substrate requires phosphorylation of

serines at the degron sequence (DSGXXS) present in the substrate protein. Does NLRP3 have such degron sequences? What is the kinase responsible for the phosphorylation of serines in the degron? YAP may inhibit this phosphorylation pathway at some point to stabilize NLRP3. This possibility must be examined.

3) Fig. 4d and 4e are very important data when considering the mechanism of the proinflammatory function by YAP, but it is a delicate data with small and subtle differences, which reduces the reliability of the paper. In particular, given that the immunoblot analysis of Fig. 4e showing the abundance of immunoprecipitated NLRP3 appears to have been performed outside of the range in which quantification is clearly maintained, a small increase in the abundance of β -TrCP bound to NLRP3 cannot be justified by a single experiment. For both experiments, multiple experiments should be performed to adequately quantify the bands using statistical validation.

4) YAP is regulated in various ways by signals from upstream kinases (LATS, etc.). In this paper, such upstream involvement is completely ignored, but what happens to this proinflammatory function of YAP when LATS is activated or silenced?

Minor comments:

1) The authors are completely mistaken about the use of the term “SCF/beta-TrCP.” SCF/beta-TrCP refers to the entire complex of four proteins: Skp1, Cull1, Rbx1, and beta-TrCP, and descriptions such as “FLAG-SCF/beta-TrCP” are incorrect. The readers do not understand which of these four proteins is attached with FLAG tag. From the context, it is likely that the authors have attached a FLAG tag to beta-TrCP, so it should be described as “FLAG-beta-TrCP.”

2) There are two paralogs in beta-TrCP, beta-TrCP1 (Fbxw1) and beta-TrCP2 (Fbxw11). Given that the phenotypes of the knockout mice are completely different from each other, their biological functions are thought to be very different. In this article, the molecular name is simply described as “beta-TrCP,” and there is no precise description of whether beta-TrCP1 or beta-TrCP2 was used in the experiments. Unless this point is

properly described, it is not possible to conduct follow-up experiments that can be verified by a third party, and this should be improved.

3) Figure 1e: What antibody was used in the upper blot (pellet)?

Responses to comments

In accordance with the suggestions, we carefully revised the manuscript and performed several additional experiments. By doing so, we have strengthened the conclusion of our paper and the physiological relevance of our findings. The point-by-point responses to the comments were listed as below.

Major comments

1. This is a very important question. We appreciate the valuable suggestions.

To explore the mechanism by which YAP inhibits the binding between NLRP3 and β -TrCP1, three possible explanations may explain the underlying mechanisms. First, YAP might affect the serine phosphorylation of NLRP3, leading to the decreased binding between NLRP3 and β -TrCP1 (FEBS J, 2020 Oct 6;Oncogene,2004, 23(11): 2028-2036). Second, YAP might directly interact with β -TrCP1 and thereby mask the NLRP3 binding site of β -TrCP1. Third, YAP might directly interact with NLRP3 and thereby mask the β -TrCP1 binding site of NLRP3. To test the first possibility, we overexpressed YAP in iBMDMs and assessed NLRP3 serine phosphorylation. LPS stimulation led to the increase of NLRP3 serine phosphorylation, which was not affected by YAP overexpression, excluding the first possibility (**Supplementary Fig.5a in the revised version**). To test the second possibility, we overexpressed YAP-S366A mutant (in humans, the homologous mutant is YAP-S381A), which lost the β -TrCP1 binding capacity (**Supplementary Fig.5b in the revised version**). However, the S366A substitution in YAP did not affect the NLRP3 and β -TrCP1 binding (**Supplementary Fig.5c in the revised version**), suggesting that YAP interrupts the NLRP3 and β -TrCP1 interaction independent of its direct binding to β -TrCP1. Next, we tested the third possibility and observed an association between YAP and NLRP3 (**Supplementary Fig.5d in the revised version**). This phenomenon was confirmed by using Proximity Ligation Assay (PLA), which could visualize the protein-protein interactions *in vivo* (**Supplementary Fig.5e, f in the revised version**). As shown by co-immunoprecipitation assay, we found that the C-terminal transactivation domain (residues 151-488) rather than the N-terminal TEAD binding domain of YAP physically interacted with NLRP3 (**Supplementary Fig.5g, h in the revised version**). Co-

expression of wild-type YAP or its C-terminal transactivation domain but not the N-terminal TEAD-binding domain interrupted the interaction between NLRP3 and β -TrCP1 (**Supplementary Fig.5i in the revised version**), indicating that the NLRP3-YAP binding inhibits the NLRP3- β -TrCP1 interaction. Further, we observed that both YAP and β -TrCP1 bound to the same domain of NLRP3 (NACHT and LRR domain) (**Supplementary Fig.5j, k, l in the revised version**). Taken together, these results demonstrate that YAP disrupts the interaction between NLRP3 and β -TrCP1 through competing with β -TrCP1 to bind NLRP3.

2. According to the suggestions, we scanned the NLRP3 protein sequence and identified two potential phosphodegron motifs, namely 193-DSPMSS-198 and 890-NSGLTS-895, respectively. Given that serine phosphorylation within the degron is crucial for its recognition by β -TrCP (FEBS J, 2020 Oct 6; Oncogene,2004, 23(11): 2028-2036), we generated NLRP3 mutants in which all serine residues within the putative degron motifs were replaced with alanine. We found that NLRP3 S891A/S895A mutant lost the ability to bind β -TrCP1, while NLRP3 S194A/S197A/S198A had similar binding capacity to β -TrCP1 as wild-type NLRP3 (**Supplementary Fig.4e, f in the revised version**), suggesting that the 890-NSGLTS-895 motif of NLRP3 is crucial for its binding to β -TrCP1.

As suggested, we checked the function of YAP in serine phosphorylation of NLRP3, and observed an increased serine phosphorylation of NLRP3 upon LPS stimulation. However, YAP overexpression did not affect the serine phosphorylation of NLRP3 (**Supplementary Fig.5a in the revised version**), so we did not further explore which kinase is responsible for phosphorylation of serine on NLRP3. We added these contents to the discussion section in the revised manuscript.

3. We performed additional experiments and showed another representative results in the modified version of the manuscript. Meanwhile, we displayed quantitative analysis of protein levels in the bottom panel in revised figure (**Fig.5d, e in the revised version**).

4. We accepted the valuable suggestions and determined whether cellular stresses-

induced Hippo signaling could affect the NLRP3 inflammasome activation, and subjected cells to serum starvation or high cell confluence, both of which are well-known cellular stress conditions that activate the Hippo signaling (Genes Dev. 2012, 26, 2138-2143; Cell, 2012, 150, 780-791). This led to YAP Ser127 phosphorylation and YAP degradation, indicative of the activation of Hippo signaling (**Supplementary Fig.3a, b in the revised version**). Serum starvation or high cell confluence markedly inhibited the NLRP3 inflammasome-induced release of IL-1 β and IL-18 in WT but not YAP-deficient macrophages, without affecting the secretion of TNF- α (**Fig.3a, b in the revised version**). Moreover, silencing of Lats1/2 expression in macrophages prevented the decrease of YAP expression (**Supplementary Fig.3c, d in the revised version**), and enhanced NLRP3 agonists-induced IL-1 β and IL-18 secretion during serum starvation (**Fig.3c in the revised version**). Thus, our findings indicated that the Hippo signaling suppresses the NLRP3 inflammasome activation in a YAP-dependent manner.

Minor comments:

1. We apologize for the mistake and corrected the term SCF ^{β -TrCP} to β -TrCP1 in modified version of the manuscript
2. We appreciated for the valuable suggestion. In our study, β -TrCP is β -TrCP1 (Fbxw1), so we made a description in revised manuscript.
3. We thank the referee for pointing it out. The antibody used in the upper blot of Figure 1e is ASC, we added it to the revised manuscript.

Reviewer #3 (Remarks to the Author):

This manuscript describes the regulation of NLRP3 by K27 ubiquitination by the E3 ligase, SCF β -TrCP. Yap maintains stability of NLRP3 by blocking its interaction with the E3 ligase and preventing K27 ubiquitination and degradation of NLRP3. Overall, the findings are clear and conclusions are well supported by the data. I had some minor comments.

Minor comments:

Spelling errors and grammatical errors throughout – should be carefully screened.

1. Figure 3c

Authors transfect a range of ubiquitin mutants into cells and examine the ubiquitination of NLRP3.

Only the K27 mutant appears to ubiquitinate NLRP3. This is surprising as the authors state that NLRP3 can undergo K48 and K63 ubiquitination, yet experimentally, they do not appear to have validated this. The authors should comment on this result. The authors convincingly show that YAP reduces K27 ubiquitination. Furthermore, the authors conclude that YAP does not prevent ubiquitination at any other lysine residue. This claim is invalid as transfection of HEK cells with any other ubiquitination mutant did not result in an increase in ubiquitination.

2. Figures 3a, 3c (lane 1) and 4a

The HA input IB shows banding in all lanes at 130 kDa even those not designated as containing transfected HA tagged proteins. I was unable to determine whether the authors intended to indicate that the HA tag was an alternative molecular weight. If this is the case, the correct HA band needs to be made clearer. Alternatively, this may be a labelling error and this needs to be corrected. Otherwise this brings into question the validity of all these IPs.

3. Overall mechanism:

Authors state that YAP interacts with SCF β -TrCP based on previous references (Beginning of results section titled Yap suppresses E3 ligase SCF β -TrCP). And from their diagram (Figure 7) the authors indicate a direct interaction of YAP with SCF β -TrCP, however they have not showed any data demonstrating a direct interaction

between these two proteins. An alternative to this is binding to 14-3-3 – a known interacting partner for Yap, which has also been shown to bind to pyrin (Jeru et al *ARTHRITIS & RHEUMATISM* Vol. 52, No. 6, June 2005, pp 1848–1857). This is something should be commented on.

Responses to comments

In accordance with the suggestions, we carefully revised the manuscript and performed several additional experiments. By doing so, we have strengthened the conclusion of our paper and the physiological relevance of our findings. The point-by-point responses to the comments were listed as below.

Minor comments

The English expression and any grammatical mistake have been carefully checked. And we revised the manuscript accordingly.

Major comments

1. The lane of K48 and K63 ubiquitination of NLRP3 were not clear might due to the shorter exposure time. We repeated the experiments and increased the exposure time. Indeed, we observed that NLRP3 can undergo K48 and K63 ubiquitination, while overexpression of YAP did not affect it (**Fig.4c in the revised version**). For other ubiquitin mutants, we fixed the wording issue and replaced “prevent ubiquitination at any other lysine residue” with “YAP had no appreciable effect on the NLRP3 ubiquitination of other types of linkages” in the revised version of manuscript.

2. This is a labeling error. We apologize for the mistake. We repeated the experiments and provided another representative results in the revised manuscript (**Fig.4a, c,5a in the revised version**).

3. This is a very important question. We appreciate the valuable suggestions.

To explore the mechanism by which YAP inhibits the binding between NLRP3 and β -TrCP1, three possible explanations may explain the underlying mechanisms. First, YAP might affect the serine phosphorylation of NLRP3, leading to the decreased binding between NLRP3 and β -TrCP1 (FEBS J, 2020 Oct 6; Oncogene, 2004, 23(11): 2028-2036). Second, YAP might directly interact with β -TrCP1 and thereby mask the NLRP3 binding site of β -TrCP1. Third, YAP might directly interact with NLRP3 and

thereby mask the β -TrCP1 binding site of NLRP3. To test the first possibility, we overexpressed YAP in iBMDMs and assessed NLRP3 serine phosphorylation. LPS stimulation led to the increase of NLRP3 serine phosphorylation, which was not affected by YAP overexpression, excluding the first possibility (**Supplementary Fig.5a in the revised version**). To test the second possibility, we overexpressed YAP-S366A mutant (in humans, the homologous mutant is YAP-S381A), which lost the β -TrCP1 binding capacity (**Supplementary Fig.5b in the revised version**). However, the S366A substitution in YAP did not affect the NLRP3 and β -TrCP1 binding (**Supplementary Fig.5c in the revised version**), suggesting that YAP interrupts the NLRP3 and β -TrCP1 interaction independent of its direct binding to β -TrCP1. Next, we tested the third possibility and observed an association between YAP and NLRP3 (**Supplementary Fig.5d in the revised version**). This phenomenon was confirmed by using Proximity Ligation Assay (PLA), which could visualize the protein-protein interactions *in vivo* (**Supplementary Fig.5e, f in the revised version**). As shown by co-immunoprecipitation assay, we found that the C-terminal transactivation domain (residues 151-488) rather than the N-terminal TEAD binding domain of YAP physically interacted with NLRP3 (**Supplementary Fig.5g, h in the revised version**). Co-expression of wild-type YAP or its C-terminal transactivation domain but not the N-terminal TEAD-binding domain interrupted the interaction between NLRP3 and β -TrCP1 (**Supplementary Fig.5i in the revised version**), indicating that the NLRP3-YAP binding inhibits the NLRP3- β -TrCP1 interaction. Further, we observed that both YAP and β -TrCP1 bound to the same domain of NLRP3 (NACHT and LRR domain) (**Supplementary Fig.5j, k, l in the revised version**). Taken together, these results demonstrate that YAP disrupts the interaction between NLRP3 and β -TrCP1 through competing with β -TrCP1 to bind NLRP3.

We accepted the valuable suggestion. As 14-3-3 σ is reported to promote the ubiquitination of c-myc or COP1 (EMBOJ, 2006, 25(6):1196-206; Nat Commun, 2015,16;6:7530), we investigated the role of 14-3-3 σ in NLRP3 ubiquitination. However, co-expression of 14-3-3 σ did not enhance the ubiquitination levels of NLRP3 (**Supplementary Fig.4a in the revised version**). We have discussed the

function of 14-3-3 family in the revised manuscript.

REVIEWERS' COMMENTS

Reviewer #1 (Remarks to the Author):

The authors have added new data that provides more mechanistic insight into how YAP regulates NLRP3. They also provide new data and a more robust discussion that clearly demonstrates the biological significance of their findings. They have addressed all of my comments and I support publication of this revised manuscript.

Reviewer #2 (Remarks to the Author):

The revision has been done very well, and I have no further points.

Reviewer #3 (Remarks to the Author):

Overall, the authors clearly show that a YAP B-TrCP1 axis mediates the stability of the NLRP3 protein and hence the level of NLRP3 inflammasome activation. They provide convincing biologically relevant links for how this axis may impact human disease including atherosclerosis. The recent additional data to demonstrate the mechanism through which Yap inhibits B-TrCP1 mediated ubiquitination of NLRP3 are interesting, well thought out and demonstrate clear molecular mechanisms.

Minor comments

Supplemental Figure 3b, the authors have shown that non-specific increasing cell densities increase YAP phosphorylation, in corresponding Figure 4b, they have shown the effect of increasing cell density in terms of number of cells plated in 6, 12 or 24 wells. For clarity, it would be helpful to keep labelling consistent. i.e to label the immunoblot in Sup. Figure 3 with corresponding cell plating densities.

Supplemental Figure 3e, immunoblot is not always the best way rule out expression of a protein in a system, as levels of protein expression may be below the limit of detection of the chemiluminescent substrate. qPCR data to show expression levels of TAZ in myeloid cells compared to lung would provide a more sensitive mechanism of detection.

Supplemental Figure 4d, both NLRP3 and B-TrCP1 are highly expressed in the cell. The immunofluorescent image does not convincingly show co-localisation of these proteins given that they are so highly and widely expressed. However, given that their IP provided in Supplemental figure 4b does demonstrate an interaction between NLRP3 and B-TrCP1, I believe those data are acceptable.

Reviewer #3 (Remarks to the Author):

Overall, the authors clearly show that a YAP B-TrCP1 axis mediates the stability of the NLRP3 protein and hence the level of NLRP3 inflammasome activation. They provide convincing biologically relevant links for how this axis may impact human disease including atherosclerosis. The recent additional data to demonstrate the mechanism through which Yap inhibits B-TrCP1 mediated ubiquitination of NLRP3 are interesting, well thought out and demonstrate clear molecular mechanisms.

Minor comments

Supplemental Figure 3b, the authors have shown that non-specific increasing cell densities increase YAP phosphorylation, in corresponding Figure 4b, they have shown the effect of increasing cell density in terms of number of cells plated in 6, 12 or 24 wells. For clarity, it would be helpful to keep labelling consistent. i.e to label the immunoblot in Sup. Figure 3 with corresponding cell plating densities.

Supplemental Figure 3e, immunoblot is not always the best way rule out expression of a protein in a system, as levels of protein expression may be below the limit of detection of the chemiluminescent substrate. qPCR data to show expression levels of TAZ in myeloid cells compared to lung would provide a more sensitive mechanism of detection.

Supplemental Figure 4d, both NLRP3 and B-TrCP1 are highly expressed in the cell. The immunofluorescent image does not convincingly show co-localisation of these proteins given that they are so highly and widely expressed. However, given that their IP provided in Supplemental figure 4b does demonstrate an interaction between NLRP3 and B-TrCP1, I believe those data are acceptable.

Responses to comments

Minor

According to the comments, we corrected the labelling in Supplemental Figure 3b

We appreciated the valuable suggestion and measured the mRNA expression of TAZ by qPCR. As shown in **Supplementary Fig.3f**, TAZ expression is much lower in primary macrophages than in lung tissues, consistent with the western-blot analysis showing protein expression of TAZ.

We agreed that immunoprecipitation (IP) is more convincing to prove the interaction between NLRP3 and β -TrCP1